# Global contributions of mesoscale dynamics to meridional heat transport

Andrew Delman[1] and Tong Lee[1]

[1]Jet Propulsion Laboratory, California Institute of Technology, Pasadena, CA, USA

**Correspondence:** Andrew Delman (adelman@jpl.caltech.edu)

**Abstract.**

Mesoscale ocean processes are prevalent in many parts of the global oceans, and may contribute substantially to the meridional movement of heat. Yet earlier global surveys of meridional temperature fluxes and heat transport (HT) have not formally distinguished between mesoscale and large-scale contributions, or have defined eddy contributions based on temporal rather than spatial characteristics. This work uses spatial filtering methods to separate large-scale (gyre and planetary wave) contributions from mesoscale (eddy, recirculation, and tropical instability wave) contributions to meridional HT. Overall, the mesoscale temperature flux (MTF) produces a net poleward meridional HT at mid-latitudes and equatorward meridional HT in the tropics, thereby resulting in a net divergence of heat from the subtropics. In addition to MTF generated by propagating eddies and tropical instability waves, MTF is also produced by stationary recirculations near energetic western boundary currents, where the temperature difference between the boundary current and its recirculation produces the MTF. The mesoscale contribution to meridional HT yields substantially different results from temporally-based "eddy" contributions to meridional HT, with the latter including large-scale gyre and planetary wave motions at low latitudes. Mesoscale temperature fluxes contribute the most to interannual and decadal variability of meridional HT in the Southern Ocean, the tropical Indo-Pacific, and the North Atlantic. Surface eddy kinetic energy (EKE) alone is not a good proxy for MTF variability in regions with the highest time-mean EKE, though it does explain much of the temperature flux variability in regions of modest time-mean EKE. This approach to quantifying mesoscale fluxes can be used to improve parameterizations of mesoscale effects in coarse-resolution models, and assess regional impacts of mesoscale eddies and recirculations on tracer fluxes.

## 1   Introduction

In regions of the ocean where waters of different temperatures and densities converge, instabilities form that transport heat across latitude lines. Many of these instabilities (eddies) assume scales comparable to or slightly larger than the baroclinic radius of deformation (e.g., Chelton et al., 2011), corresponding to the oceanic mesoscale (tens to hundreds of kilometers). In addition to eddies, tropical instability waves (TIWs; e.g., Jochum and Murtugudde, 2006) and recirculation gyres that flank

boundary current jets also have the capacity to move heat meridionally in the ocean at mesoscales. Unlike the wind-forced

response associated with larger-scale gyres and planetary waves, these mesoscale phenomena are generated and sustained by nonlinear mechanisms such as baroclinic and barotropic instability (e.g., Eady, 1949; Charney and Stern, 1962) and nonlinear momentum advection (e.g., Greatbatch et al., 2010).

Despite the prevalence of mesoscale features in the global oceans, relatively little attention has been given to quantifying their contributions to meridional heat transport (HT) until recently. Most of the literature characterizing the oceanic meridional

HT emphasizes the dominant role of the overturning circulation (e.g., Talley, 2003), arising from the steep vertical temperature gradients at lower latitudes. Some of these studies assess the "gyre" contribution separately from the overturning (e.g., Bryan, 1982; Hall et al., 2004; Johns et al., 2011). In this framework, the overturning contribution is the integrated product of the zonal mean meridional velocity $v$ and temperature $T$, and the gyre contribution consists of the residual (zonally-varying) $v$ and $T$. However, no distinction is made between the temperature flux associated with the basin-scale gyres vs. smaller mesoscale

processes. This distinction is important not only because the forcing mechanisms for basin-scale motions are quite different from mesoscale motions, but also because the coarse-resolution oceans in most climate model simulations do not explicitly represent mesoscale motions and must parameterize their effects. In recent years, eddy tracking and identification methods (e.g., Chaigneau and Pizarro, 2005; Chelton et al., 2011; Laxenaire et al., 2018) have been used to quantify the specific meridional HT or temperature flux contributions of identified eddies (e.g., Hausmann and Czaja, 2012; Dong et al., 2014, 2017; Sun et al.,

2019; Müller et al., 2019; Laxenaire et al., 2020). Yet not all mesoscale flow features take the form of coherent vortices, and the actual movement of coherent eddies likely accounts for a relatively small portion of the fluxes associated with the mesoscale (e.g., Hausmann and Czaja, 2012; Abernathey and Haller, 2018; Sun et al., 2019).

Another commonly-used approach is to quantify the "eddy" contribution to meridional HT based on the deviation of $v$ and $T$ from the temporal (rather than zonal) mean (e.g., Cox, 1985; Jayne and Marotzke, 2002; Aoki et al., 2013; Griffies et al.,

2015; Ushakov and Ibrayev, 2018). Jayne and Marotzke (2002) used this formulation to assess eddy temperature fluxes and meridional HT, finding eddy temperature fluxes in energetic mid-latitude regions (Antarctic Circumpolar Current, western boundary current extensions) that are consistent with downgradient diffusivity estimates (e.g., Stammer, 1998). Volkov et al. (2008) defined the eddy contribution as the deviation from 3-month means of $v$ and $T$, hence their high-frequency eddy contribution to meridional HT was somewhat lower than the total time-varying contribution quantified by Jayne and Marotzke

(2002). Yet these temporal decomposition methods may conflate the contributions of large-scale and mesoscale circulations, as gyres and long planetary waves have temporal covariances between $v$ and $T$. Moreover, the effects of stationary mesoscale features (e.g., recirculation gyres) are not included in the temporal eddy meridional HT contributions. Since spatial resolution prevents many climate models from explicitly simulating mesoscale ocean dynamics, accurate representations of the ocean depend on parameterizations (e.g., Gent and McWilliams, 1990; Eden and Greatbatch, 2008; Marshall et al., 2012) that must

take into account stationary as well as transient mesoscale fluxes.

This study extends the methodology that Delman and Lee (2020) used in the North Atlantic, in order to better understand the specific contribution of mesoscale processes to meridional temperature fluxes and meridional HT globally. Three focus areas of this study are to: (1) assess the time-mean contributions of the meridional mesoscale temperature flux (MTF), (2) compare the

mesoscale contribution to meridional HT with other "eddy" meridional HT diagnostics, and (3) quantify interannual/decadal variability in the MTF and determine the extent to which surface eddy kinetic energy (EKE) can be used as a proxy for MTF variability on these timescales. Section 2 discusses the ocean general circulation model and methods used to quantify the components of meridional HT. Section 3 maps the mesoscale temperature flux (MTF) globally and discusses its contributions from stationary vs. time-varying (propagating) mesoscale dynamics. Section 4 summarizes mesoscale contributions to basin-integrated meridional HT, and compares mesoscale vs. temporally-based measures of the eddy contribution to meridional HT. Section 5 assesses the global distribution of MTF interannual and decadal variability and its relationship to surface EKE. Section 6 discusses the key conclusions of this study and identifies areas in need of future investigation.

## 2    Methods

### 2.1    Model simulation and assessment of EKE

To quantify mesoscale temperature fluxes and contributions to meridional HT, this analysis uses output from an eddy-permitting ocean general circulation model, the Parallel Ocean Program (POP) 2 (Smith et al., 2010). POP integrates the primitive equations with a $z$-depth coordinate, and is configured in a tripole grid with two north poles over Canada and Siberia. The simulation was run on the Yellowstone computing cluster (Computational and Information Systems Laboratory, 2016), with a 0.1° longitude grid spacing in the Mercator portion of the grid south of ∼28°N, and progressively finer grid spacing approaching the two north poles. In physical distance, the grid spacing is approximately 11 km near the equator, 5.5 km at 60°S and 5–7.5 km at 60°N. Given this spacing relative to the baroclinic deformation radius (e.g., Hallberg, 2013; Wekerle et al., 2017), it is expected that the model should at least permit mesoscale instability development equatorward of 50°-60° latitude. The model simulation has 62 depth levels with 10 meter vertical spacing in the upper 160 meters. The ocean surface is forced with Coordinated Ocean-Ice Reference Experiments version 2 (COREv2; Large and Yeager, 2009) fluxes based on National Centers for Environmental Prediction reanalysis with corrections from satellite data. The simulation was spun up during a 15-year period forced by CORE normal-year forcing (Large and Yeager, 2004), followed by a 33-year model integration with COREv2 interannually-varying fluxes corresponding to the years 1977–2009. Our analysis covers the 32-year period 1978–2009, the same span of time as in Delman and Lee (2020). For more details of the model simulation see Johnson et al. (2016) and Delman et al. (2018).

As a proxy for the model's representation of mesoscale activity globally, the model time-mean surface EKE was compared with surface EKE from altimetry data. The altimetry dataset used is produced by Collecte Localisation Satellites (Ducet et al., 2000) and made available through the Copernicus Marine Environment Monitoring Service (CMEMS), with data merged from numerous altimetry missions onto a grid at 1/4° spatial and daily temporal resolution. The analysis demonstrates that the locations and EKE levels of the most energetic regions of the ocean are well represented in the POP simulation (Figure 1). However, it is also important to note that POP underrepresents EKE in many other regions of the ocean; including the subtropical eddy bands between 15° and 30° latitude in the North and South Pacific and South Indian Oceans, and to a lesser extent the TIW region near the equator. This underestimation of EKE in POP persists even if a different filter wavelength is

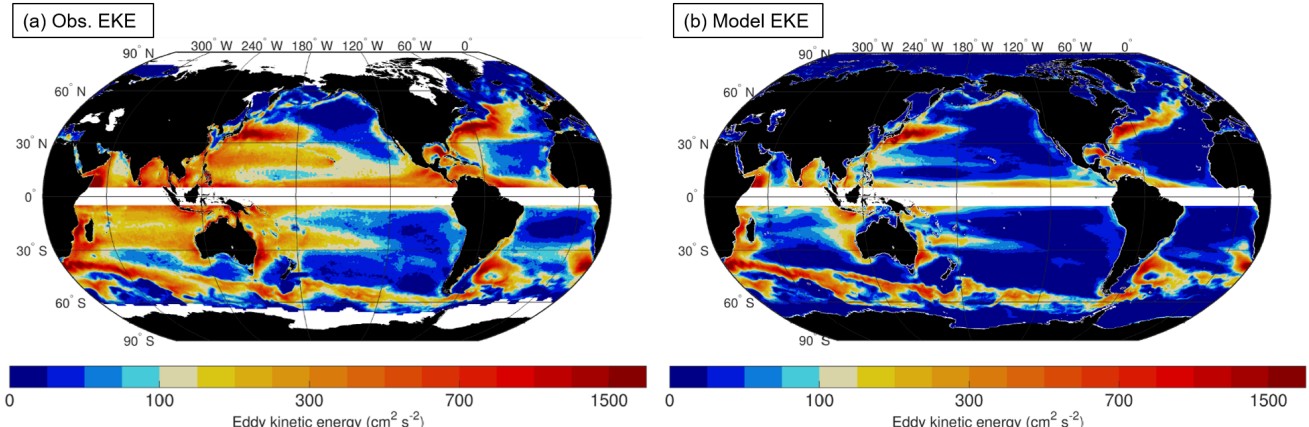

**Figure 1.** Time mean surface eddy kinetic energy (EKE), from (a) the Copernicus Marine Environment Monitoring Service (CMEMS) satellite-based gridded product, and (b) the POP model, with wavelengths <0.5° filtered out using a two-dimensional application of the filter in eq. (6). Time averages were computed 1993–2016 from the satellite data and the years corresponding to 1978–2009 in the model simulation. Regions within 5° latitude of the equator are masked out.

applied to both model and altimetry datasets (e.g. 1°), and indicates a consistent low bias of mesoscale activity in lower-energy parts of the ocean interior. One possible explanation is that the biharmonic viscosity $\nu_0$ = -2.7 x $10^{10}$ m$^4$ s$^{-1}$ and diffusivity $\kappa_0$ = -3 x $10^9$ m$^4$ s$^{-1}$ values used in this model simulation are well tuned for energetic regions such as western boundary currents, but may suppress too much mesoscale activity in the less energetic ocean interiors. However, the low bias in time-mean EKE is not unique to this eddy-permitting model (e.g., Volkov et al., 2008, 2010; Tréguier et al., 2012). An attempt to estimate the impact of this bias on mesoscale contributions to meridional temperature fluxes and heat transport is discussed in Appendix B.

## 2.2 Computing the mesoscale temperature flux

The most common method of quantifying an "eddy" flux defines the eddy meridional velocity $v'$ and temperature $T'$ as deviations from a time mean (e.g., $v = \overline{v} + v'$). The contributions to meridional HT are given by

$$\int \overline{vT}\, dx = \int \left( \overline{v}\,\overline{T} + \overline{v'T'} \right) dx \tag{1}$$

and the second term on the right-hand side is the eddy contribution. In this paper the temporal eddy flux is also referred to as the "all time-varying" flux, encompassing the effects of transient/propagating phenomena at large scales as well as mesoscales. The time mean may also be implemented on shorter timescales (an example with 3-month time averages will be discussed in Section 4), or another type of time averaging (e.g., seasonal climatological averages) may be used.

In contrast, the primary focus of this manuscript uses a decomposition that targets the mesoscale by explicitly separating spatial (rather than temporal) scales. This method was used by Delman and Lee (2020), following a similar application of spatial filters by Zhao et al. (2018), but with the additional separation of the overturning contribution and corrections to the filtered $v$ and $T$ profiles near lateral boundaries. The meridional temperature flux is decomposed into three components (overturning, large-scale, and mesoscale) by applying spatial filters to zonal profiles of $v$ and $T$. The temperature flux is related to the heat flux by a factor of $\rho c_\rho$, where $\rho$ is the density and $c_\rho$ the specific heat capacity of seawater; the heat flux integrated across a transect with near-zero net volume transport is a HT. In the spatial decomposition used in this study, the meridional temperature flux at each depth level is decomposed into zonal mean and zonal deviation components

$$\int vT\,dx = \int \left(\langle v\rangle\langle T\rangle + v''T''\right)dx \tag{2}$$

where $\langle\rangle$ indicates the zonal mean and $''$ indicates the deviation from the zonal mean. The first term on the right-hand side $\langle v\rangle\langle T\rangle$ is the overturning component of the meridional temperature flux, consistent with earlier studies (e.g., Bryan, 1982; Johns et al., 2011). The overturning component, when multiplied by $\rho c_\rho$ and vertically integrated, gives the portion of meridional HT associated with basin-wide vertical gradients in meridional flow and temperature. The second term, variously referred to as the gyre or eddy temperature flux, contains all contributions from horizontally-varying $v$ and $T$. Our method further decomposes this term into contributions from large-scale and mesoscale variations in $v$ and $T$

$$v'' = v_L + v_M \tag{3}$$

$$T'' = T_L + T_M \tag{4}$$

$$v''T'' = v_L T_L + \left[(v_L T_M + v_M T_L) + v_M T_M\right] \tag{5}$$

The first term on the right-hand side is the contribution from the large-scale circulation, while the remaining three terms constitute the mesoscale temperature flux (MTF). The cross terms $v_L T_M + v_M T_L$ are considered part of the mesoscale contribution since these fluxes would not exist without mesoscale structures. In practice, $v_L T_M$ is generally negligible compared to the other terms since zonal $T$ profiles are red-shifted (have more large-scale structure) compared to zonal $v$ profiles. However, $v_M T_L$ can be the largest contributor to the MTF, especially near boundaries where large-scale temperature deviations from the zonal mean are often substantial.

In this analysis, transects are extracted from the model output along tracks corresponding to the nearest grid faces to integer lines of latitude. In the non-Mercator portions of the grid in the Northern Hemisphere, this results in a zig-zag of the transect

through the model grid, but in this way the net volume transport through each transect is conserved. The spatial filters to compute the large-scale and mesoscale $v$ and $T$ are applied in the spectral (zonal wavenumber) domain, with $v''$ weighted by $\Delta x$ (the length of each grid face along the transect) to better conserve volume transport across the transect. The Fourier coefficients $V''(k)$ of $v''\Delta x$ and $T''$ are filtered according to the transfer functions

$$V_L(k) = \left[0.5 + 0.5\text{erf}\left(-s\,ln\frac{|k|}{k_0}\right)\right] V''(k) \tag{6}$$

$$V_M(k) = \left[0.5 + 0.5\text{erf}\left(s\,ln\frac{|k|}{k_0}\right)\right] V''(k) \tag{7}$$

The forms of the low-pass (6) and high-pass (7) filters are symmetric and contain a steepness factor $s$ that affects the rate of signal roll-off near the cutoff wavenumber; a higher value approaches a boxcar filter with associated "ringing" effects, while a lower value avoids ringing but at the expense of cutoff precision. In filtering for large-scale and mesoscale $v$ and $T$, a value of $s = 5$ was selected as an optimal balance of roll-off between the physical coordinate and spectral wavenumber domains. The sets of Fourier coefficients $V_L$ and $V_M$ resulting from the two filters sum to the original coefficients $V''$. The threshold wavenumber $k_0 = 1/\lambda_0$ is the reciprocal of the threshold wavelength $\lambda_0$, which defines the scale separation between large-scale and mesoscale. Values of $\lambda_0$ in this study were set to $10°$ longitude poleward of $20°$ latitude in both hemispheres, increasing to $20°$ longitude within $10°$ of the equator. More details and a justification of these choices are given in Appendix A.

To better preserve zero net volume/mass flux in the basin-integrated $v_L$ and $v_M$ components, our method incorporates boundary and channel corrections (described in more detail in Delman and Lee, 2020). These corrections also aim to improve local representation of the large-scale/mesoscale separation near boundaries and in narrow channels. Before filtering, (1) $v$ profiles over land areas are set to zero, and (2) a buffer is applied to temperature profiles over land areas near boundaries to avoid sharp swings in $T_L$. After filtering, (3) non-zero $v_L$ and $v_M$ that leaked onto land areas is returned to water areas, and (4) within channels bounded by bathymetry that are narrower than $\lambda_0/4$, the meridional velocity profiles are set to $v_L = v$ and $v_M = 0$. Steps (1), (3), and (4) are taken in order to improve conservation of volume along the transect, while steps (2) and (3) improve the local representation of the large-scale and mesoscale separation. Regarding the rationale for step (4), at most latitudes mesoscale activity peaks near wavelength $\lambda_0/2$ (e.g., Figure A1), so $\lambda_0/4$ is approximately the diameter of a typical mesoscale eddy. Channels narrower than $\lambda_0/4$ are therefore too narrow to support typical mesoscale instabilities, but transport in these channels can contribute substantially to the large-scale circulation (e.g., Indonesian Throughflow, Gulf Stream in the Florida Strait). Hence all of the transport in these narrow channels has been assigned to the large-scale component.

Lastly, as with the total temperature flux, the time-mean MTF may contain contributions from the time-mean $v$ and $T$, as well as time-varying $v$ and $T$

$$\overline{\text{MTF}} = \overline{\text{MTF}_{\text{stat}}} + \overline{\text{MTF}_{\text{vary}}} \tag{8}$$

$$\text{MTF}_{\text{stat}} = \overline{v_L}\,\overline{T_M} + \overline{v_M}\,\overline{T_L} + \overline{v_M}\,\overline{T_M} \tag{9}$$

$$\text{MTF}_{\text{vary}} = \overline{v'_L T'_M} + \overline{v'_M T'_L} + \overline{v'_M T'_M} \tag{10}$$

with $^{-}$ indicating a time average. The mesoscale stationary flux $\text{MTF}_{\text{stat}}$ is the contribution of the time averages of $v$ and $T$, and is often associated with boundary current recirculations and standing meanders. In this analysis the time averages are applied over the full 32 years of model output used. The mesoscale time-varying flux $\text{MTF}_{\text{vary}}$ is associated with rectified fluxes from transient motions (e.g., instability-generated eddies).

## 3 Global distribution of time-mean mesoscale temperature fluxes

### 3.1 Total mesoscale temperature flux

Once computed, the distribution of the MTF in the global oceans highlights where mesoscale dynamics contribute most substantially to the lateral movement of heat in the oceans. There is however an important caveat when inferring the physical significance of the MTF. Flux vectors consist of rotational and divergent fluxes (Marshall and Shutts, 1981); by definition, the rotational temperature (and heat) fluxes do not contribute to changes in ocean temperature or heat content and therefore are not generally of interest for climate studies. Our decomposition method is applied only in one horizontal dimension (zonally) so the divergent flux can not be neatly separated from the rotational flux except in basin integrals (where the rotational flux is negligible). However, Jayne and Marotzke (2002) showed that rotational fluxes in mesoscale-active areas typically take on mesoscale structure (i.e., they recirculate at mesoscales), while divergent meridional fluxes tend to have larger-scale structure particularly in the zonal direction.

Hence, in this study a zonal smoothing filter is applied to maps of temperature fluxes to reduce contamination from the "noise" of the rotational fluxes. This smoothing filter has the same form as the low-pass filter in equation (6). However, for this smoothing the threshold wavenumber $k_0$ is chosen to be half the threshold wavenumber (twice the threshold wavelength) for the large-scale/mesoscale separation at that latitude. This larger value of the threshold wavelength is used to remove more of the rotational fluxes that occur at the mesoscale. In the smoothing filter the steepness factor $s$ is also set to 2 (rather than 5), because the smoothing effect is more important here than the precision of the wavelength cutoff. The smoothed MTF indicates the effect of the mesoscale circulation rectified to larger scales. It is the impact of mesoscale fluxes on large-scale temperature distributions that is most relevant when quantifying the fluxes that need to be parameterized in coarse-resolution models.

Maps of zonally-smoothed time-mean MTF (Figure 2) show that the mesoscale contributions to meridional HT are generally concentrated near the western boundary. Even in the tropics, MTF contributions are mostly confined to the western boundaries, with the exception of the TIW band in the north-central equatorial Pacific, and to a lesser extent in the eddy-rich region south of Indonesia. The impact of MTF contributions is generally to flux heat equatorward in the tropics, and poleward at higher latitudes. However, there are exceptions where the MTF fluxes heat equatorward even at mid-latitudes, most notably in

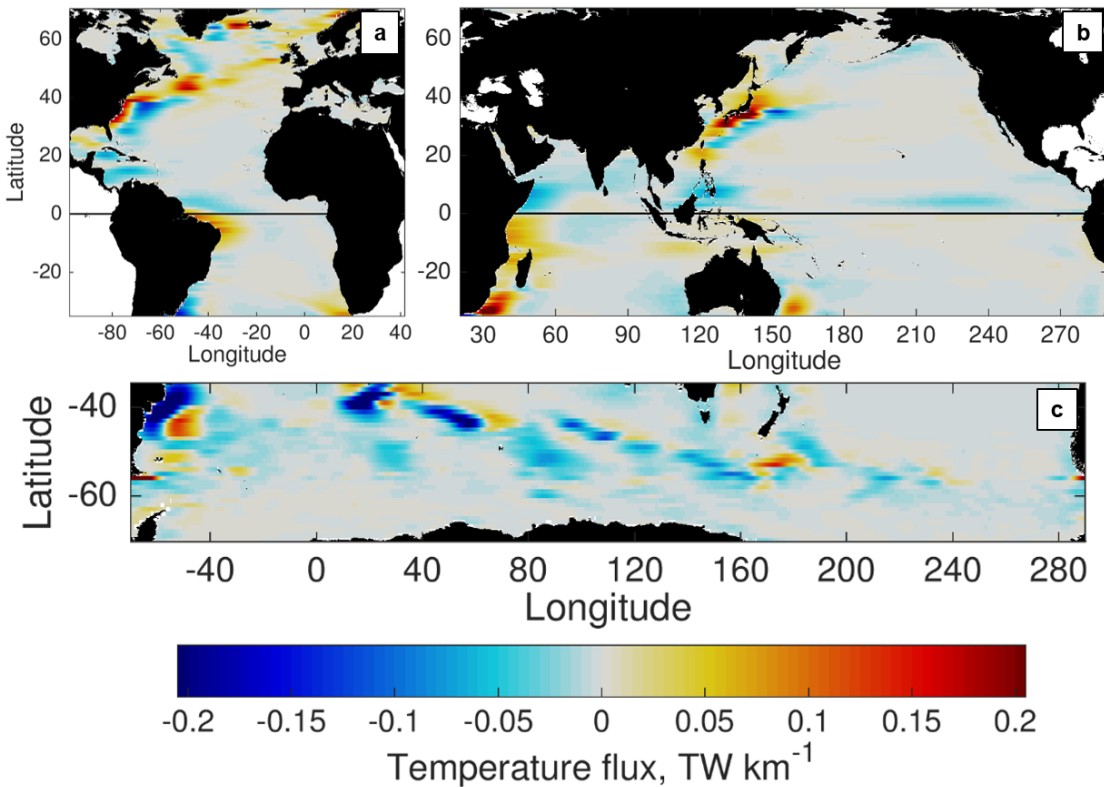

**Figure 2.** POP time-mean northward temperature flux associated with the mesoscale flow (MTF), in units of terawatts per kilometer (TW km$^{-1}$), zonally low-passed using a wavelength equal to twice the large-scale/mesoscale threshold wavelength at each latitude. Fluxes shown in the (a) Atlantic, (b) Indo-Pacific, and (c) Southern Ocean basins.

the western boundary currents of the South Indian (Agulhas Current) and South Pacific (East Australian Current) near 30°S (Figure 2b). Equatorward MTF is also apparent in the Labrador Sea at the western edge of the North Atlantic subpolar gyre (Figure 2a). We note that the significant low EKE bias in the more quiescent middle regions of the ocean (Figure 1) implies that MTF may be underestimated in these regions, especially at low latitudes. An assessment of the potential impact of this bias using composite averages is discussed in Appendix B.

### 3.2 Contributions of stationary versus time-varying mesoscale structure

The stationary part of the MTF, excluded from temporal definitions of the eddy flux, comprises a substantial portion of the total MTF globally (Figure 3). At lower latitudes (equatorward of 40° in both hemispheres), contributions from MTF$_{\mathrm{stat}}$ are overwhelmingly concentrated near the western boundaries of ocean basins. In the Southern Ocean, stationary contributions are found near meridional excursions in the Antarctic Circumpolar Current (ACC) at the Brazil-Malvinas Confluence and south of

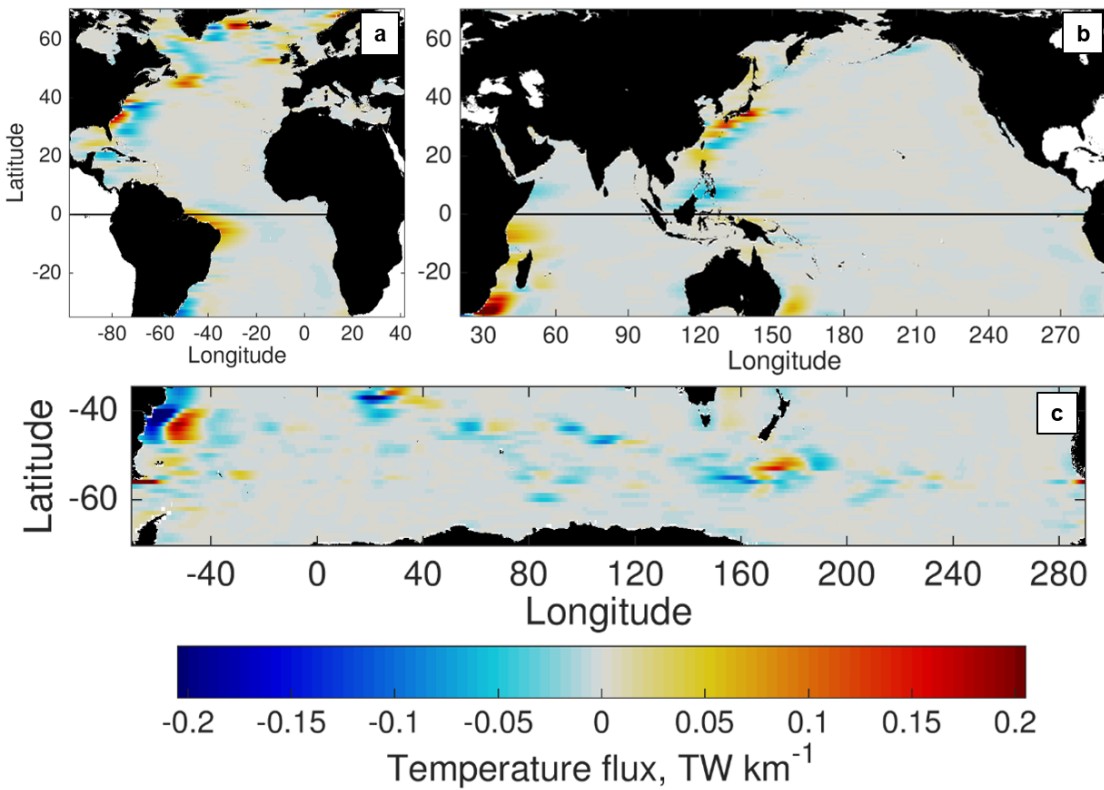

**Figure 3.** POP time mean meridional temperature flux associated with the stationary mesoscale flow only ($\overline{\mathrm{MTF_{stat}}}$), zonally low-passed. Fluxes shown in the (a) Atlantic, (b) Indo-Pacific, and (c) Southern Ocean basins.

New Zealand. In the North Atlantic, $\mathrm{MTF_{stat}}$ contributions are found in the Labrador Sea, Irminger Sea (between Greenland and Iceland), and near the coasts of the British Isles and Scandinavia. Because of the importance of the flow and temperature structure at mesoscales, all of these regions are areas where temperature fluxes are likely to be incorrectly represented in coarse-resolution climate models, unless these fluxes are properly parameterized.

The time-varying $\mathrm{MTF_{vary}}$ is closer to the temporal eddy flux $v'T'$, but our definition excludes larger-scale temporal variability. As might be expected, $\mathrm{MTF_{vary}}$ contributions are found mostly in regions where the model also has relatively high EKE (Figure 4): the Gulf Stream and North Atlantic Current, the Kuroshio Extension, the north equatorial Pacific, South Equatorial Current in the Indian Ocean, and the ACC. Many areas that have large $\mathrm{MTF_{stat}}$ also have large $\mathrm{MTF_{vary}}$, including most western boundary regions and the ACC. However, ocean interior regions such as the North Atlantic Current and the north

equatorial Pacific TIW area have large $\mathrm{MTF_{vary}}$ without substantial $\mathrm{MTF_{stat}}$ contributions, indicating that the flows driving these fluxes are both mesoscale and transient. In the Southern Ocean, the Agulhas Return Current (near 40°S, 0–60°E) and the lee of the Kerguelen Plateau (near 50°S, 80–90°E) are areas where the MTF is dominated by the transient rather than

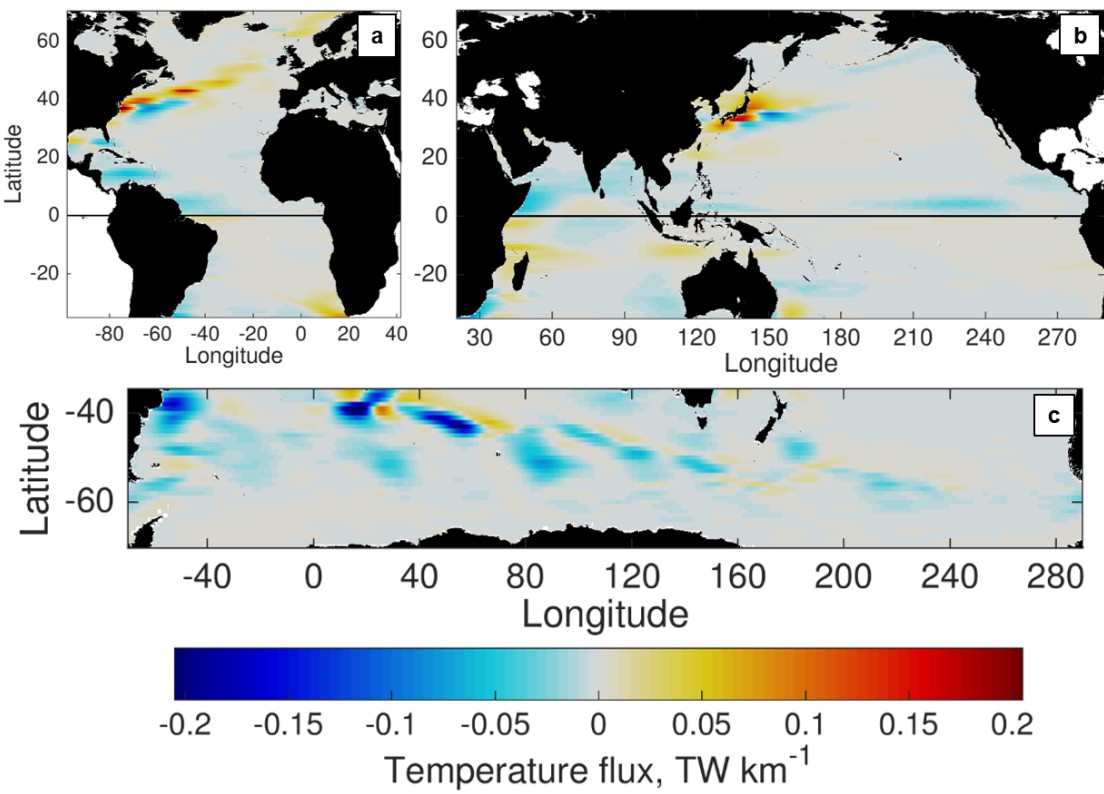

**Figure 4.** POP time mean meridional temperature flux associated with the time-varying mesoscale flow only ($\overline{\mathrm{MTF}_{\mathrm{vary}}}$), zonally low-passed. Fluxes shown in the (a) Atlantic, (b) Indo-Pacific, and (c) Southern Ocean basins.

the stationary contribution. The distribution of $\mathrm{MTF}_{\mathrm{vary}}$ in Figure 4 resembles the divergent component of the temporal eddy flux (Jayne and Marotzke, 2002; Aoki et al., 2013), but with some distinctions; for instance, the near-equatorial fluxes are

220 not as high-amplitude in Figure 4. This may be due to the dominance of long planetary waves at these latitudes, which are time-varying phenomena (temporal "eddy") but our analysis considers to be large scale, not mesoscale.

Some insight can be gained about the contributions of stationary vs. time-varying contributions to meridional HT by considering the cumulative zonal integral of each MTF component. Figure 5 shows this analysis for latitudes in the Indo-Pacific with relatively large MTF. It can be seen that stationary MTF contributions are typically associated with western boundaries

(the western Pacific at 32°N, 4°N, and 13°S, and the western Indian at 32°S), and that the zonally-integrated $\mathrm{MTF}_{\mathrm{stat}}$ contribution is much larger than $\mathrm{MTF}_{\mathrm{vary}}$ at the mid-latitude locations. Time-varying $\mathrm{MTF}_{\mathrm{vary}}$ contributions are more substantial at the low-latitude locations (4°N and 13°S), and can be found both near the basin boundaries (Somali Current and Mozambique Channel/Mascarene Basin respectively) and in the ocean interiors (Pacific TIWs and South Equatorial Current eddies respectively).

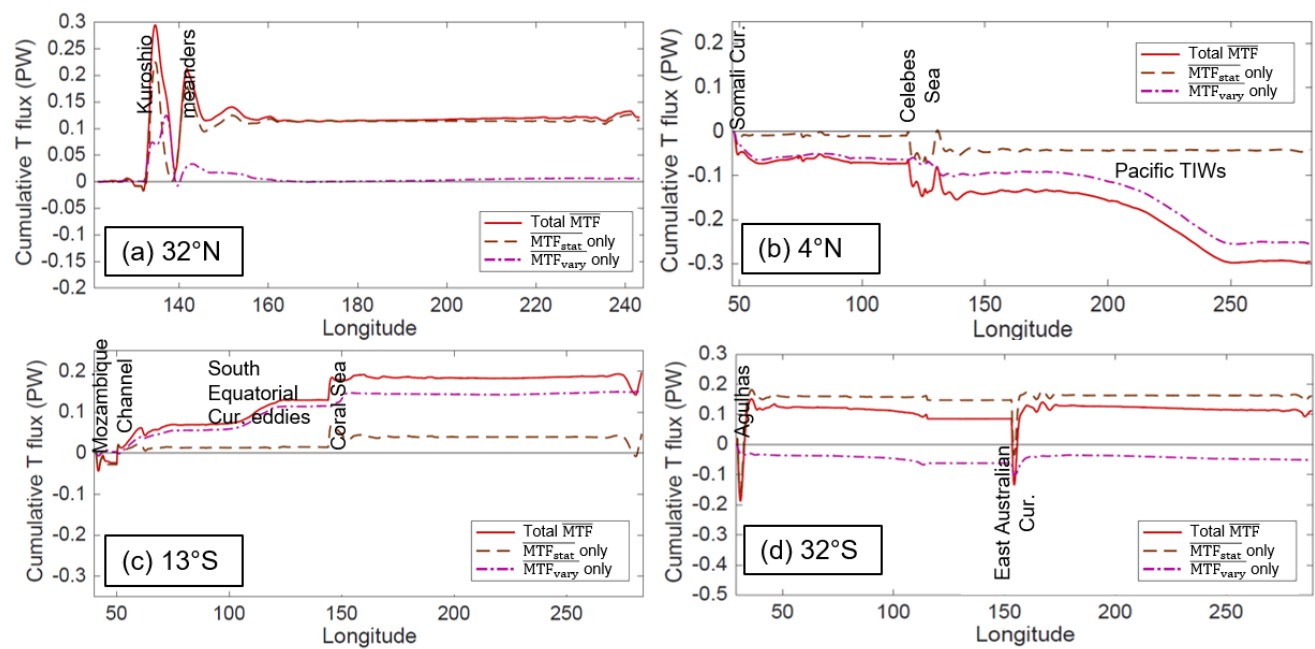

**Figure 5.** The time mean full-depth mesoscale temperature flux in the Indo-Pacific basin, cumulatively integrated from west to east, showing the total MTF, the part of the MTF attributed to time mean $v$ and $T$ components only $\overline{\mathrm{MTF}_{\mathrm{stat}}}$, and the part attributed to time-varying $v$ and $T$ components that have a rectified contribution to time-mean MTF $\overline{\mathrm{MTF}_{\mathrm{vary}}}$. Transects are shown for latitudes (a) $32°$N, (b) $4°$N, (c) $13°$S, and (d) $32°$S.

230  Since stationary mesoscale contributions to meridional HT are substantial (Figure 5), it is important to understand the velocity and temperature structure that contributes to the flux. For example, the $\mathrm{MTF}_{\mathrm{stat}}$ contributions near mid-latitude western boundaries are driven by a temperature difference between the poleward WBC jet and the equatorward recirculation (Figure 6). In the Gulf Stream, Kuroshio, and Brazil Current poleward of $\sim 30°$ latitude, the poleward jet is associated with a local maximum in temperature due to the advection of waters from lower latitudes (Figure 6a-c). As a result, the water in the

235 jet is warmer than the water in the recirculation, and the net MTF is poleward (Figure 2a-b). In contrast, the jet of the Agulhas is coincident with a steep gradient in temperature without a local maximum in temperature (Figure 6d); hence the water in the jet is cooler than in the recirculation and the net MTF is equatorward. The sign of the MTF associated with WBCs therefore depends on whether the local temperature profile is determined more by the temperature advection of the WBC jet (poleward MTF), or the frontal gradient that the WBC is aligned with (equatorward MTF). Most WBCs have both characteristics, but the

240 degree to which one is dominant can be the difference between a stationary flux that is 0.12 PW poleward (Figure 5a) and 0.16 PW equatorward (Figure 5d).

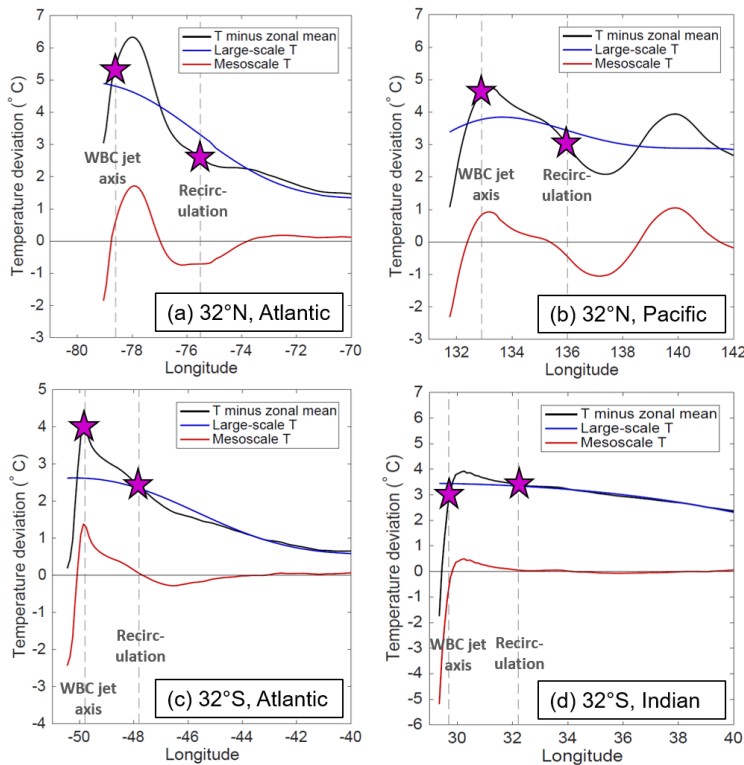

**Figure 6.** Time mean temperature profiles at 95 m depth near the western boundary in the specified latitude and basin. The vertical gray dashed lines indicate the locations of peak velocities associated with the poleward western boundary current (WBC) jet, and the equatorward recirculation. The difference in temperature between these two locations (magenta stars) provides an indication of the mesoscale temperature flux from the jet and recirculation.

## 4 Contributions to zonally-integrated meridional HT

### 4.1 Spatial decomposition of meridional HT

When integrated zonally across the global ocean or within ocean basins, the contribution of the mesoscale temperature flux
(MTF) to time-mean meridional HT is largest at mid-latitudes, with substantial contributions in the Indo-Pacific tropics as well
(Figure 7). The largest magnitude mesoscale contribution to meridional HT at any latitude is at 40°S, with a poleward heat
transport (-0.6 PW) powered by strong MTF in the Brazil-Malvinas Confluence and Agulhas Return Current regions (Figure
2). This exceeds the contribution of the large-scale meridional HT and nearly counters the equatorward meridional HT (+0.8
PW) associated with the overturning at that latitude. The Northern Hemisphere mid-latitude meridional HT contributions are
not as large; however, the mesoscale meridional HT contribution in the North Atlantic at 43°–45°N is almost comparable to the
overturning at those latitudes (in this case both components are poleward). In the North Pacific, the mesoscale contributes to

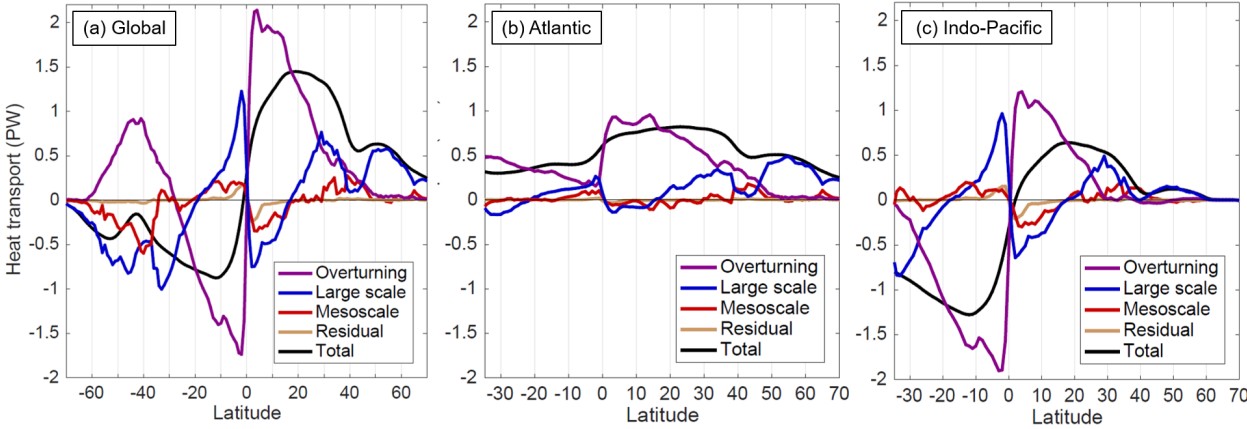

**Figure 7.** Spatial decomposition of the time mean meridional heat transport as a function of latitude (a) globally, and in the (b) Atlantic and (c) Indo-Pacific. Positive values indicate a northward heat transport. The residual consists of the effect of high-frequency co-variations in $v$ and $T$ (timescales <5 days) archived as fluxes in POP, which could not be spatially decomposed into individual $v$ and $T$ profiles.

poleward meridional HT at ~35°–42°N where the overturning and large-scale contributions essentially vanish. Both the Southern and Northern Hemisphere mid-latitude peaks in mesoscale meridional HT coincide with local minima in the magnitude of the large-scale meridional HT (Figure 7a). This suggests that the mesoscale plays an important part in conveying poleward meridional HT from the subtropical to subpolar gyres in the Northern Hemisphere near 40°N, and across the equatorward edges of the Southern Ocean near 40°S. Additional notable mesoscale contributions to time-mean meridional HT are found at the Indo-Pacific at 2°-9°N (-0.3 PW) and at 15°-11°S (+0.2 PW).

A key focus of this study is a better understanding of the interannual/decadal (ID) variability of the mesoscale contribution to meridional HT; accordingly the time series of the spatial components at each latitude have been temporally low-passed for periods >14 months and the seasonal cycle explicitly removed. The standard deviations of the ID filtered components (Figure 8) show that the overturning dominates meridional HT variability at most latitudes. This is not surprising since the overturning component of meridional HT highlights the effect of temperature differences between the shallow and deep ocean, and at lower latitudes these vertical differences are much larger than zonal temperature differences across ocean basins. However, mesoscale contributions to meridional HT variability can be comparable to the overturning and large-scale contributions at mid and high latitudes. Globally, the amplitude of mesoscale meridional HT variability at ID timescales is generally comparable to the large-scale variability poleward of 30° latitude, and comparable to the overturning variability poleward of 40° latitude in both hemispheres (Figure 8a). The largest peaks in ID mesoscale meridional HT variability occur at 43°–40°S, 3°–4°N, and 32°N, with the latter two peaks explained mostly by contributions from the Indo-Pacific (Figure 8c). These latitudes correspond to the high EKE regions associated with the Agulhas Return Current, Pacific TIWs, and Kuroshio respectively, implying that the very active mesoscale dynamics and interannual variability in these areas result in large MTF variability on ID timescales as well.

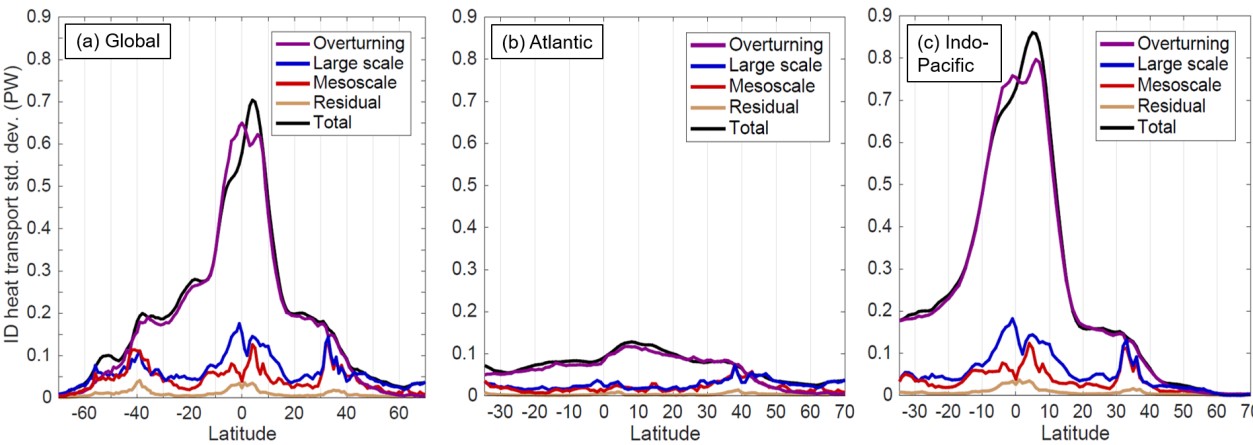

**Figure 8.** Interannual/decadal (ID) standard deviation of the spatial components of meridional heat transport (a) globally and in the (b) Atlantic and (c) Indo-Pacific basins. The ID time series of each component of meridional HT have been low-passed for periods longer than 14 months, with the seasonal cycle explicitly removed.

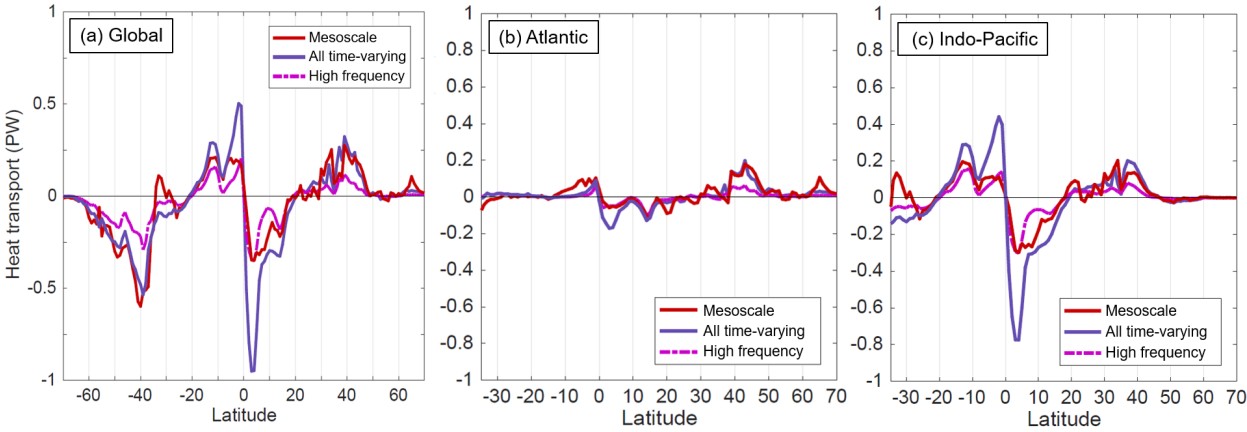

**Figure 9.** Comparison of various "eddy" formulations of the meridional heat transport as a function of latitude (a) globally, and in the (b) Atlantic and (c) Indo-Pacific. The three formulations are the mesoscale component computed based on eq. (5), the all time-varying component $\overline{v'T'}$, and the high frequency component computed from $v'$ and $T'$ on timescales shorter than 3 months.

The mesoscale contribution to meridional HT using our method can also be compared to two definitions of the eddy contribution based on a temporal decomposition. The "all time-varying" contribution consists of the residual eddy term on the right-hand side of eq. (1) when the contribution from the 32-year time averages of $v$ and $T$ are removed. The "high frequency" contribution, following (Volkov et al., 2008), is the residual eddy term when time averages of $v$ and $T$ in each 3-month period are removed (i.e., consists of variability at timescales shorter than $\sim$3 months). When zonally-integrated globally and across

ocean basins, neither the all time-varying nor high frequency components are uniformly consistent with the mesoscale contribution (Figure 9). Generally, the mesoscale meridional HT is closer to the high frequency contribution at low latitudes, and closer to the all time-varying contribution at mid latitudes. One possible inference from this is that mesoscale dynamics tend to have consistently high frequencies in the tropics (where planetary wave and eddy propagation speeds are higher), while mesoscale dynamics at higher latitudes tend to propagate more slowly and take on lower frequencies. The difference between all time-varying and mesoscale meridional HT close to the equator may also be associated with the impact of long Kelvin and Rossby waves, which are time-varying with frequencies ranging from intraseasonal to interannual, but with spatial wavelengths comparable to the size of an entire ocean basin (e.g. Boulanger and Menkès, 1999; McPhaden and Yu, 1999). Hence the temporal eddy fluxes conflate the contributions of large-scale planetary wave activity and mesoscale eddy activity. In contrast, the mesoscale component isolates the contributions to meridional HT from fluxes that are not expected to be well represented in coarse-resolution climate models.

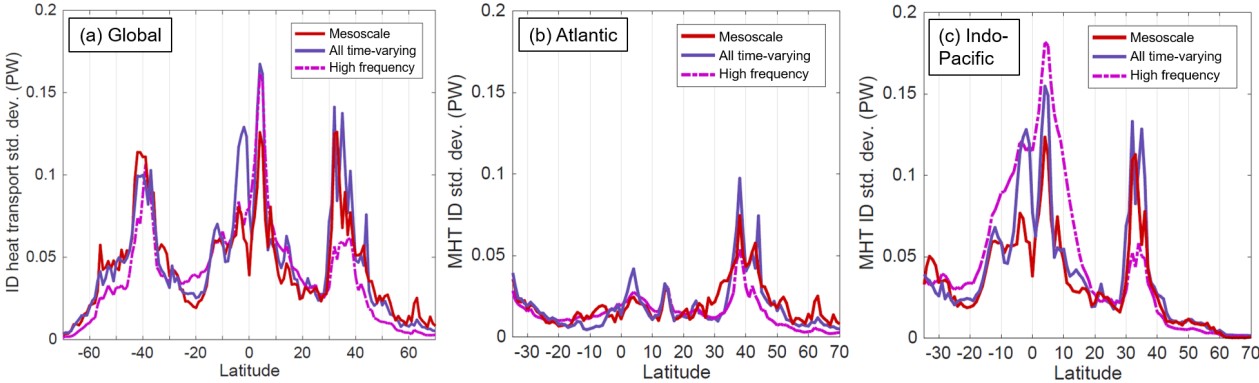

**Figure 10.** Interannual/decadal (ID) standard deviation of the spatial and temporal "eddy" formulations (a) globally and in the (b) Atlantic and (c) Indo-Pacific basins.

A comparison of the standard deviation associated with the ID time series of the three components (Figure 10) shows that all three components have spikes in temporal variability in the mid-latitudes and near the equator. As with the contributions to time-mean meridional HT, the ID variability of the mesoscale contribution more closely resembles the time-varying contribution at mid-latitudes. In the tropics, the ID variability of the mesoscale does not resemble either component; in the tropical Indo-Pacific the high frequency and time-varying standard deviations are both larger than the mesoscale, implying substantial contributions from large-scale ID variability.

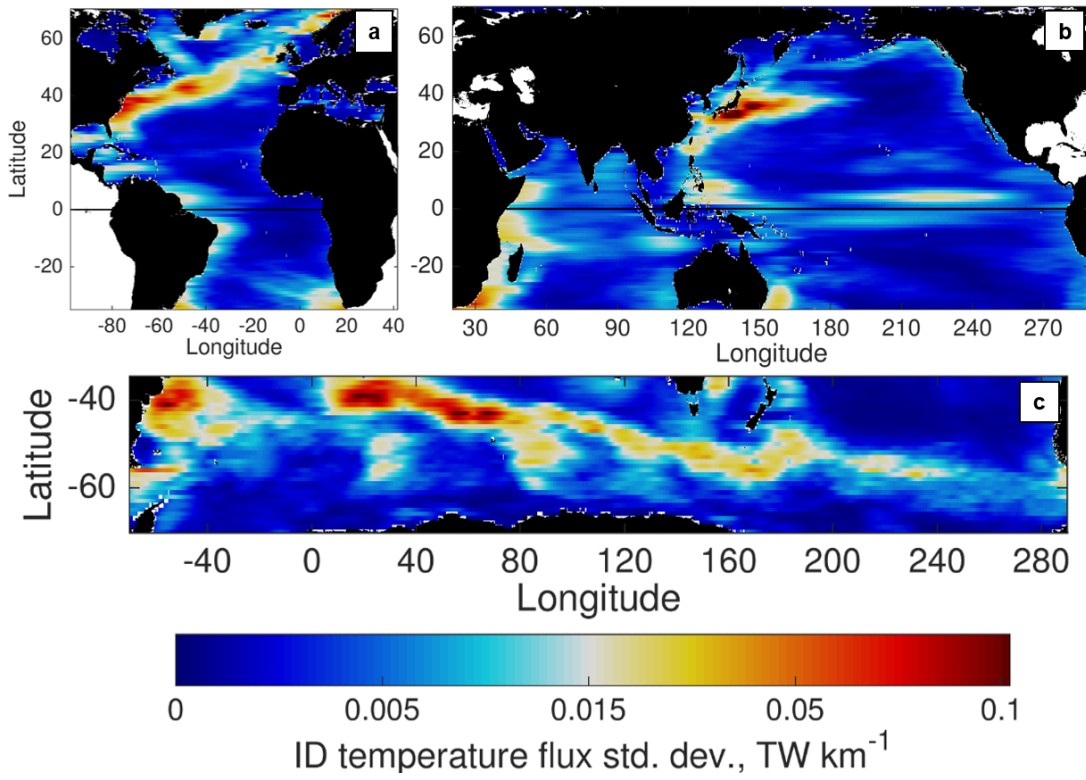

**Figure 11.** Standard deviation on ID timescales of the MTF, zonally smoothed using the same smoothing filter as in Figures 2–4. The ID time series of MTF have been low-passed for periods >14 months prior to the computation of the standard deviation. Fluxes are shown in the (a) Atlantic, (b) Indo-Pacific, and (c) Southern Ocean basins.

## 5  Mesoscale interannual/decadal variability

### 5.1  Locations of substantial mesoscale contributions

Mesoscale dynamics influence not only the time-mean meridional HT but also its variability on ID timescales (Figure 8), motivating a study of the regions where the mesoscale contributions to ID temperature flux variability are greatest. As with the time-mean MTF, the interannual and decadal variability of the MTF is mostly concentrated near western boundaries, the ACC, and the north equatorial Pacific and southern tropical Indian oceans (Figure 11). High levels of ID variability of the MTF also generally coincide with regions of high EKE in the POP model simulation (Figure 1b). Notably, the subtropical eddy bands in the Pacific and Indian Oceans seem to have a negligible impact on MTF ID variability. The POP model does underestimate EKE in these eddy bands relative to altimetry (Figure 1), but it still shows these bands as regions of elevated EKE; in contrast there is almost no elevation in ID MTF variability associated with these eddy bands. This indicates that mesoscale fluxes are not particularly efficient at moving heat meridionally in subtropical eddy bands, at least in the POP simulation.

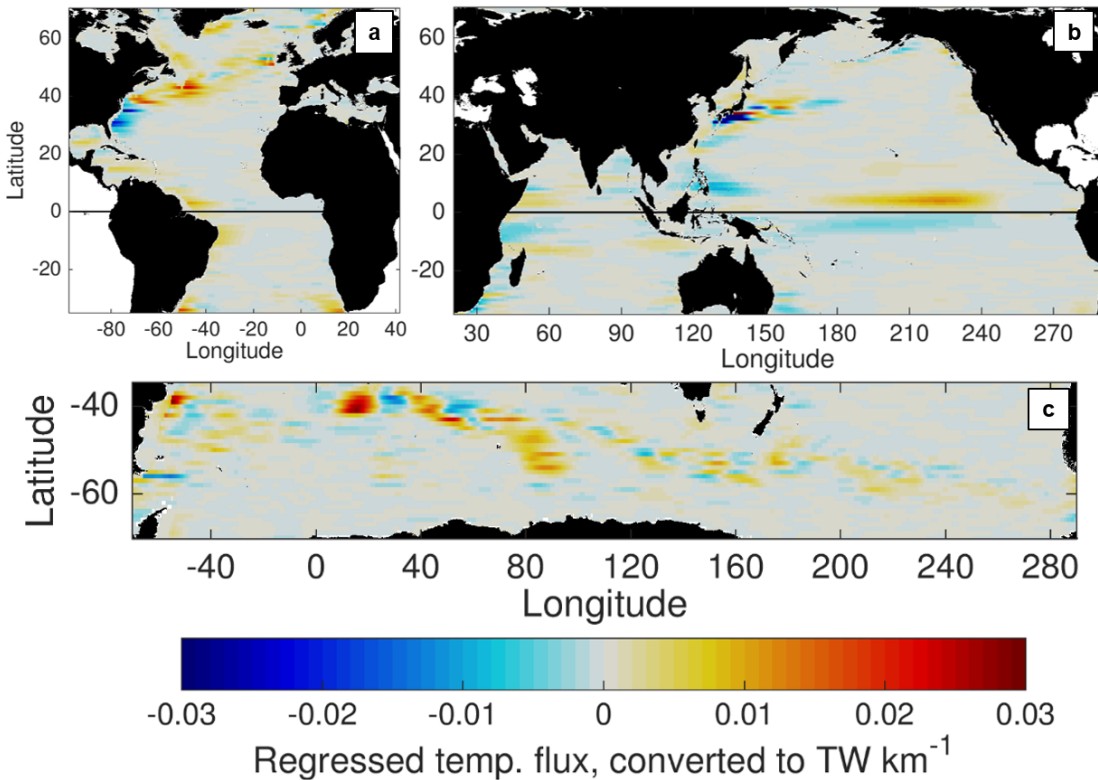

**Figure 12.** Local (zonally-smoothed) mesoscale temperature flux $R_{\mathrm{MTF}}$ that contributes to 1 standard deviation of total meridional HT variability on ID timescales, computed by linear regression at each latitude and basin. Positive values indicate that the MTF is positively correlated with (contributes to) total ID meridional HT variability; negative values indicate that the MTF is negatively correlated with (compensates) total ID meridional HT variability.

Another way to assess the contributions of MTF variability to basin-integrated meridional HT on ID timescales is to compute a linear regression or correlation between (1) the time series of meridional HT integrated across the basin at each latitude and (2) the local zonally-smoothed MTF time series along the transect. This regression-based flux contribution can be expressed as

$$R_{\mathrm{MTF}} = R_{\frac{\mathrm{loc}}{\mathrm{tot}}} \sigma_{\mathrm{tot}} = C(\mathrm{loc},\mathrm{tot})\sigma_{\mathrm{loc}} \tag{11}$$

where $R_{\frac{\mathrm{loc}}{\mathrm{tot}}}$ is the linear regression coefficient of the local MTF given the time series of the total basin-integrated temperature

flux, $\sigma_{\mathrm{loc}}$ and $\sigma_{\mathrm{tot}}$ are the standard deviations of the local and total basin-integrated time series respectively, and $C(\mathrm{loc},\mathrm{tot})$ is the correlation coefficient of the local and total basin-integrated time series. The regression flux contribution $R_{\mathrm{MTF}}$ is the local temperature flux that would be expected to contribute to a basin-integrated value of 1 standard deviation above the mean; positive (negative) values of $R_{\mathrm{MTF}}$ indicate a positive (negative) correlation of the two time series. $R_{\mathrm{MTF}}$ can be expressed in

units of temperature flux or heat flux, and the latter are used here. A similar analysis was shown in Figures 7-8 of Delman and
315 Lee (2020) in profiles of longitude-depth, but the MTF is vertically coherent in most regions, so here the "local" time series
are depth-integrated.

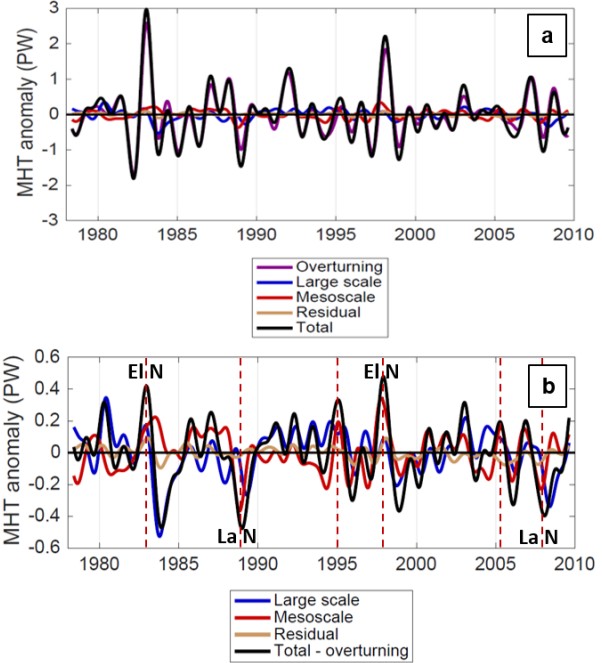

**Figure 13.** (a) Time series of spatial components of meridional HT variability in the Indo-Pacific basin across $4°$N, with time means removed.
(b) Same as (a), but with the overturning component of meridional HT variability removed. Vertical red dashed lines indicate episodes where
mesoscale MHT drives variations in the total (non-overturning) MHT. Major El Niño (El N) and La Niña (La N) events coincident with
mesoscale extrema are also annotated.

In Figure 12, $R_{\mathrm{MTF}}$ is computed such that the local time series is the MTF and the total time series is the total meridional
HT (sum of all components) at ID timescales with the mean and seasonal cycle removed. These maps show the regions where
local MTF has substantial variability that is coherent with total meridional HT variability in a given basin. Overall, three areas
emerge where the MTF variability contributes substantially (and positively) to basin-integrated meridional HT variability:
the North Atlantic, the tropical Indo-Pacific, and the Southern Ocean between $0°–100°$E. In the North Atlantic, the positive
contributions peak at $\sim43°$N, where the Grand Banks of Newfoundland protrude into the North Atlantic Current (Figure 12a).
In contrast, the "negative" contributions of the Gulf Stream and the Kuroshio south of their separations from the continental
shelf imply that MTF variations compensate other components (specifically large-scale temperature fluxes). For instance, if the
325 large-scale flow is less efficient than usual at advecting heat poleward (perhaps due to the orientation or intensity of the main
current), mesoscale variability may take up more of this flux instead. The tropical Indo-Pacific contributions are highest in the

TIW bands (Figure 12b), with a notable asymmetry across the equator: north of the equator the MTF contributes directly to meridional HT variability, while south of the equator the MTF compensates meridional HT variability from other components. In the north equatorial TIW band, the mesoscale contribution to meridional HT contributes to the total meridional HT, often in phase with the much larger overturning contribution (Figure 13); this variability is also closely related to the El Niño-Southern Oscillation. The mesoscale contributions near 13°S (Figure 13a,b) are focused in the Indian Ocean on both the western and eastern sides, though they are weaker than the near-equatorial contributions. In the Southern Ocean, the Agulhas Return Current (45°-40°S, 10°-20°E) and lee of the Kerguelen Plateau (55°-45°S, 70°-100°E) emerge as hotspots for mesoscale contributions to total meridional HT (Figure 12c). Among the above areas, there are two regions where the local MTF and basin-integrated total meridional HT are also highly positively correlated with $C(\mathrm{loc}, \mathrm{tot}) > 0.5$: the Pacific TIW region north of the equator, and the lee of the Kerguelen Plateau. The implication is that mesoscale processes in these two regions have a direct impact on the meridional HT at their respective latitudes, without much compensation from the overturning or large-scale contributions.

## 5.2 Surface EKE as a proxy for ID MTF variability

Local velocity variability and EKE have been considered in previous studies as possible observational proxies for lateral heat fluxes induced by mesoscale eddies (e.g., Stammer, 1998; Bolton et al., 2019; Müller and Melnichenko, 2020). Delman and Lee (2020) explored the degree to which variations in surface EKE can account for MTF variability at ID timescales, along the 40°N transect in the North Atlantic. By applying mixing length theory (e.g., Green, 1970; Holloway, 1986; Stammer, 1998) the relationship can be expressed as

$$\mathrm{MTF} \approx -\kappa \frac{\partial T}{\partial y} \tag{12}$$

$$\kappa \propto \sqrt{v'^2} L_{\mathrm{mix}} \approx \sqrt{\mathrm{EKE}} L_{\mathrm{mix}} \tag{13}$$

where $\propto$ indicates proportionality (not equality), $\kappa$ is the diffusivity (which is generally positive or downgradient), and $L_{\mathrm{mix}}$ is the mixing length. (The approximation in eq. 13 assumes that $u'^2 \approx v'^2$, which is valid where dynamics are generally isotropic, but not where flows are strongly asymmetric and nearly aligned with the zonal or meridional axis.) According to this construct, if the mixing length and meridional temperature gradient do not vary greatly, EKE should be correlated locally with the MTF. In this analysis, the zonal smoothing filter used on MTF in previous figures is applied to both EKE and MTF in order to reduce the impact of shifts in the location of currents and focus on the regional relationship between EKE and MTF.

Figure 14 shows the fraction of MTF variance at ID timescales explained by local variations in the surface EKE; the fraction of variance is equivalent to the squared correlation coefficient of the surface EKE and MTF. Many regions in the ocean have a significant correlation and fraction of variance explained. These areas include the southern excursions of the ACC, the subtropical bands of eddy activity in the Indo-Pacific at ~20°-25° latitude in both hemispheres, and the fringes of energetic

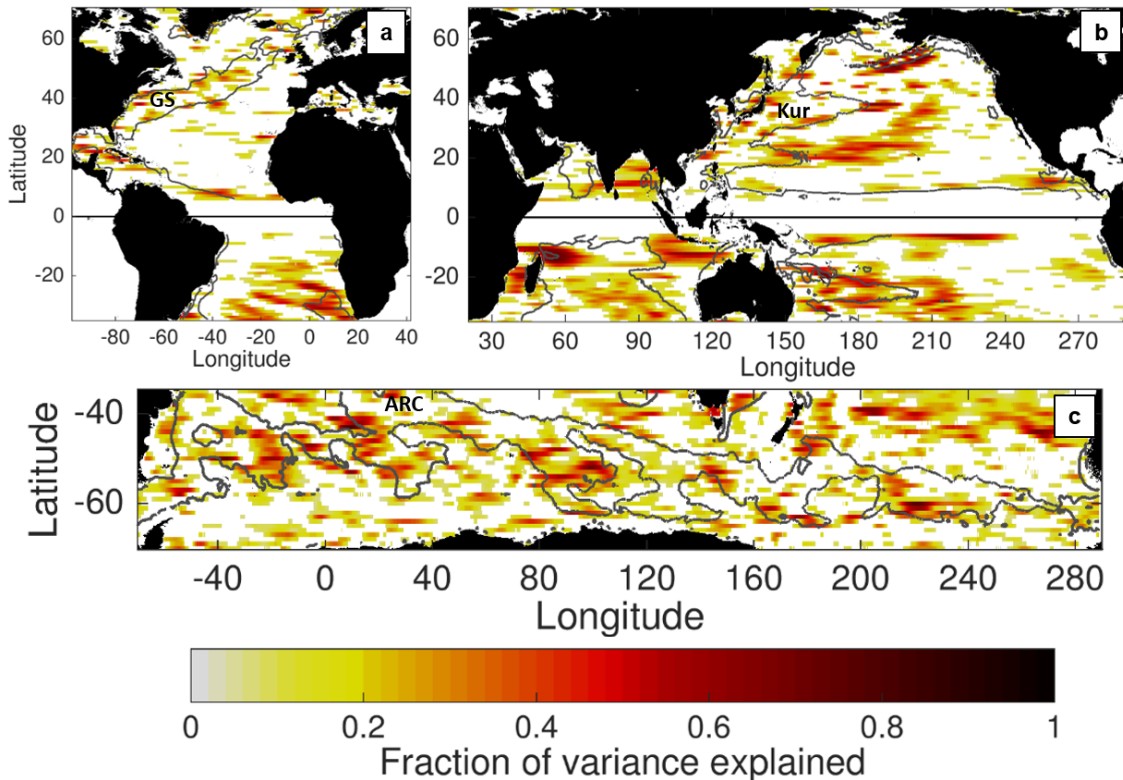

**Figure 14.** Fraction of zonally-smoothed mesoscale temperature flux variance explained by the regression onto surface EKE, at ID timescales. Only areas with a signal-to-noise ratio $> 1$ at the 95% confidence level are shaded. The gray contour indicates $100 \; \mathrm{cm}^2 \; \mathrm{s}^{-2}$ time-mean EKE in the POP model. Several regions (GS, Kur, ARC) with very high time-mean EKE are labeled.

western boundary currents. One characteristic shared by these areas is that they tend to have modest levels of time-mean EKE overall (Figure 1); in fact, many of these areas occur near the time-mean contour of $100 \; \mathrm{cm}^2 \; \mathrm{s}^{-2}$ in the POP model, which often separates energetic and quiescent regions in the ocean.

However, in the core of more energetic regions with high EKE, surface EKE is not a good proxy for the MTF. These energetic regions include the Gulf Stream and Kuroshio extensions as well as the Agulhas Return Current, which are annotated in Figure 14. The lack of significant MTF variance associated with EKE in these high-energy regions may be attributed to several factors. EKE is a convenient and widely-understood metric of mesoscale activity, but it is based on removing a temporal mean from the velocity field and it is not the most precise way to gauge the level of mesoscale energy present. Higher levels of EKE

may be associated with an anomalous placement of the current jet and associated front, which does not necessarily change the cross-jet flux. Moreover, strong velocity jets tend to suppress diffusivity across fronts by reducing the mixing length (Ferrari and Nikurashin, 2010); the suppression effect of the flow is pronounced in many of the same energetic regions where EKE is a poor proxy for MTF (Klocker and Abernathey, 2014; Groeskamp et al., 2020). Ultimately the MTF depends not only on

the amplitude of mesoscale velocity and temperature variation, but on the local correlation between $v_M$ and $T_M$ (Delman and Lee, 2020), which in turn is a measure of the efficiency of mesoscale motions in producing a net directional flux. The factors affecting this $v_M$-$T_M$ correlation locally need to be investigated more in future work. Figure 14 does however provide an indication of areas of the ocean where surface EKE might be a decent proxy for the MTF, in contrast with the more energetic regions where mixing length and gradient variability can interfere with the EKE-MTF relationship.

## 6 Conclusions

This analysis uses the method first applied in Delman and Lee (2020) in the North Atlantic to quantify mesoscale temperature fluxes in the global oceans, and their contribution to the time-mean and ID variability of meridional HT. The resulting assessment indicates that mesoscale ocean dynamics generally spread heat outward from the subtropics: poleward in mid-latitudes, and equatorward in the tropics. Hence the net effect of the mesoscale meridional HT is to flux heat from where thermocline isotherms are deepest (in subtropical gyres) to where they are shallower, consistent with the "slumping" associated with typical parameterizations of mesoscale effects in models (e.g., Gent and McWilliams, 1990). Despite this, mesoscale temperature fluxes do not always act to diminish large-scale horizontal temperature gradients. In the Agulhas and East Australian Current, mesoscale recirculations act to flux heat equatorward at mid-latitudes (Figure 3b), such that the net mesoscale meridional HT is equatorward at 35°-30°S in the Indo-Pacific (Figure 7c). Moreover, the time variability of mesoscale temperature fluxes on ID timescales is not well explained by surface EKE variability in the most energetic regions of the ocean; hence a more nuanced understanding of how mesoscale processes flux heat across strong currents is needed.

It is important to highlight the differences between the definition of the mesoscale flux described here and the "eddy" fluxes used in other studies, as described in Section 4. While eddy fluxes are typically associated with deviations from a zonal or temporal mean, neither of these definitions separates the effects of mesoscale phenomena from those of basin scale gyres or long planetary waves. Moreover, stationary mesoscale recirculations associated with western boundary currents contribute substantially to mesoscale meridional HT (e.g., the Kuroshio and Agulhas as shown in Figure 5). The effects of stationary recirculations are not included in a temporally-defined eddy flux, as these recirculations are part of the time-mean circulation. However, a coarse-resolution model that does not accurately simulate temperature and velocity gradients at western boundaries (Figure 6) may neglect contributions to the meridional HT that are >0.1 PW, even if the model accurately represents the large-scale flow and mass transport. Therefore, the effects of these stationary recirculations still need to be parameterized just as the effects of transient eddies are parameterized. By implementing this diagnostic to separate scales in mesoscale-permitting and mesoscale-resolving simulations, it is possible to better understand how stationary and transient mesoscale phenomena induce fluxes across large-scale gradients. Future studies can focus on relating large-scale gradients (in one, two, and three dimensions) to mesoscale diffusivities, and therefore on improving parameterizations of mesoscale impacts from stationary recirculations as well as propagating eddies.

The scope of this study leaves a number of questions unanswered about the effects of mesoscale dynamics on heat pathways through the ocean. While POP simulates EKE well in the Southern Ocean and western boundary currents where MTF is

substantial, the model has a low EKE bias in some subtropical and tropical regions (Figure 1). Hence the spatial decomposition method should be applied in models that simulate mesoscale activity and variability more accurately in these regions. The large-scale/mesoscale decomposition used in this study is applied only in one dimension (zonally), and therefore does not
separate divergent from rotational fluxes. Only the divergent fluxes contribute to changes in ocean heat content locally, so a separation of the large-scale and mesoscale in both horizontal dimensions is needed to compute the divergence and assess the impact of mesoscale dynamics on local heat content. Lastly, the disconnect between surface EKE and MTF variability in the most active mesoscale regions also needs further attention, as it may result from factors such as variability in large-scale temperature gradients, mixing length, and the local geometry of the mesoscale flow field. Assessing the influence of these
factors on MTF variability, perhaps with the aid of more complex statistics and/or deep learning techniques (George et al., 2019), can be expected to produce improvements in predictions of the fluxes associated with mesoscale variability.

*Data availability.* The POP model output used in this study is stored on NCAR's High Performance Storage System (HPSS); the full model output in 5-day averages is available with a user account (through https://www2.cisl.ucar.edu) by logging into cheyenne.ucar.edu and accessing the following path on HPSS: /home/bryan/johnsonb/g.e01.GIAF.T62_t12.003/ocn/hist/. Source code to run the POP2 model is
available at http://www.cesm.ucar.edu/models/cesm1.0/pop2/. The CMEMS surface dynamic topography data used to produce the analysis in Figure 1 are available from http://marine.copernicus.eu/services-portfolio/access-to-products/ by searching for the Product ID SEALEVEL_GLO_PHY_L4_REP_OBSERVATIONS_008_047.

## Appendix A: Mesoscale transition scale $\lambda_0$

Delman and Lee (2020) used $\lambda_0 = 10°$ longitude as the threshold wavelength in the North Atlantic, based on spectral analyses
of meridional velocity and temperature at a range of latitudes. For this study, similar spectral analyses were carried out in the Atlantic, Indo-Pacific, and Southern ocean basins; the meridional velocity spectra are shown at various latitudes in the Indo-Pacific and Southern ocean basins as examples (Figure A1). As in Delman and Lee (2020), the mesoscale/large-scale transition and mesoscale peak wavelengths were estimated from each transect based on the logarithmically-smoothed spectral density profiles. The choice of $\lambda_0 = 10°$ longitude was found to be reasonable at and poleward of $20°$ latitude. Even where the
objectively-identified transition wavelength was very different from $10°$ (e.g., $20°$S in the Indo-Pacific as shown in Fig. A1e), the mesoscale "bump" in the spectra clearly occurs at scales smaller than $10°$. The one exception to this pattern is at some tropical latitudes (e.g., $10°$S Indo-Pacific, Fig. A1d), where the mesoscale bump extends to scales larger than $10°$ longitude. Given this, and the abrupt growth of the deformation radius and typical eddy scales equatorward of $20°$ (Chelton et al., 2011), the transition wavelength was set to $\lambda_0 = 20°$ longitude within $10°$ of the equator, $\lambda_0 = 10°$ poleward of $20°$ latitude in both
hemispheres, with a linear transition in $\lambda_0$ in the $10°$–$20°$ latitude range.

The wavelength (not radius) $\lambda_0$ was deliberately chosen to be larger than the size of nearly all eddies, in order to ensure that their signals are retained in the mesoscale. For instance, an eddy radius of $1°$ longitude and diameter of $2°$ is typical at mid-latitudes (Chelton et al., 2011), corresponding to a wavelength of $4°$ longitude. If $\lambda_0$ at mid-latitudes is chosen to be $4°$ or

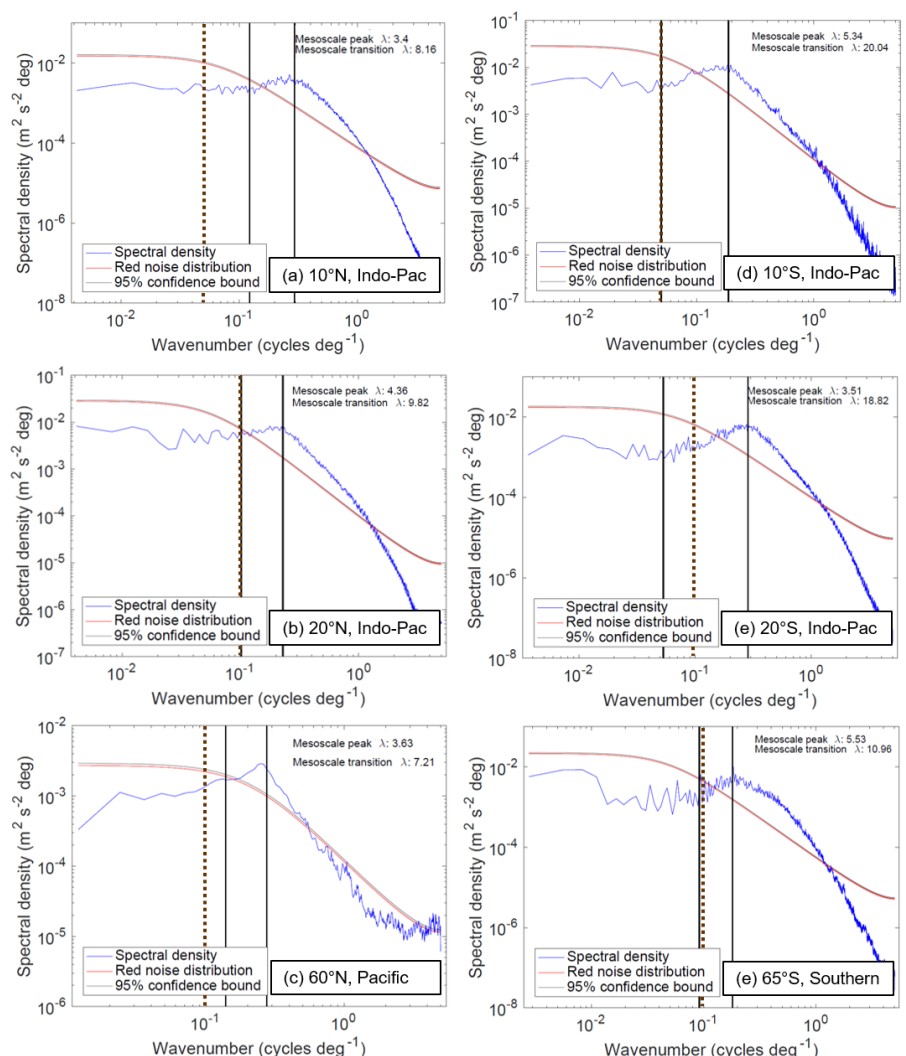

**Figure A1.** Zonal wavenumber spectral density of meridional velocity in POP, along latitude transects in specified basins. The black vertical lines and small text indicate the wavelength associated with the mesoscale transition and peak, as computed using the logarithmically-smoothed spectral density profile. The vertical dashed brown line indicates the wavelength $\lambda_0$ chosen for the mesoscale/large-scale threshold at that latitude. The red and gray curves indicate the red noise distribution expected from lag-1 (model grid-scale) autocorrelation and the corresponding 95% confidence bound.

5° longitude, much of the signal of larger eddies will be included in the large-scale profiles along with basin gyres and long planetary waves. Hence a choice of $\lambda_0 = 10°$ results in a cleaner separation of these phenomena.


## Appendix B:  Composite adjustment for model EKE bias

The POP model's significant underrepresentation of EKE (Figure 1) in some parts of the ocean is a potential concern for the results of this analysis, in that it may also lead to an incorrect representation of the fluxes and transport associated with oceanic mesoscale activity. This concern is mitigated somewhat by the fact that the low EKE biases tend to occur in calmer interior regions of the ocean where satellite observations agree that EKE is generally lower overall (Figure 1a). Nonetheless, there are areas where observed mesoscale activity is large enough that this bias might impact estimates of meridional HT, especially in the subtropical eddy bands of the Pacific and Indian oceans, and the eddy activity near the Azores Current ($\sim$35°N) in the Atlantic.

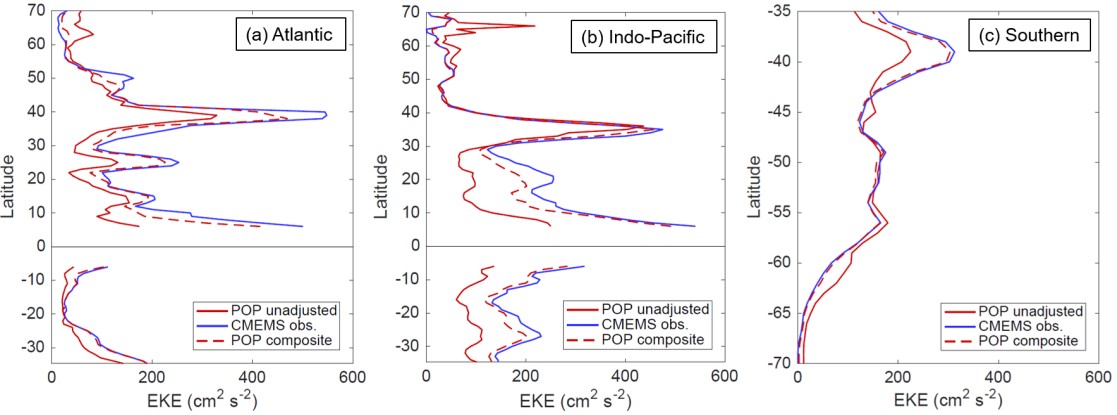

**Figure B1.** Zonally-averaged surface EKE, in the (a) Atlantic, (b) Indo-Pacific, and (c) Southern ocean basins. The three curves indicate the values from the unadjusted POP simulation, from the CMEMS satellite-based gridded data product, and from the EKE-adjusted composite that is intended to emulate EKE levels in the CMEMS observations.

An imperfect way to get an estimate of the EKE bias's impact on MTF and meridional HT contributions is to construct composite averages of MTF for a given location, in such a way that the model EKE averaged during the composite times is as close as possible to the observed time-mean EKE in that location. In areas where there is a low EKE bias in the model, these adjusted composites exclude times when the EKE is lower, in order to average MTF only at times when EKE levels are closer to observations. This is done using values of EKE that are zonally smoothed (in the same way MTF values were smoothed in Figure 2) in order to avoid the method being impacted by minor shifts in current position. One caveat to this approach is that the bias in parts of the subtropics is substantial enough that the observational EKE can not be matched exactly without limiting the composite averages to one or two anomalous events in the model. In this analysis the time range included in each adjusted composite was required to include at least 3 years cumulatively (with any number of gaps), to limit the influence of individual events or eddies. Even with this limitation, the adjusted composite EKE levels are still at least 50% of the observational time-mean in the same location for 92% of the ocean's area studied. Furthermore, the zonally-averaged composite EKE in each basin

is closer to observational values than to the unadjusted time-mean POP value (Figure B1), everywhere except for a couple of latitudes in the North Atlantic (34°N due to the bias in the Azores Current, 50°N due to the bias in the Northwest Corner of the North Atlantic Current).

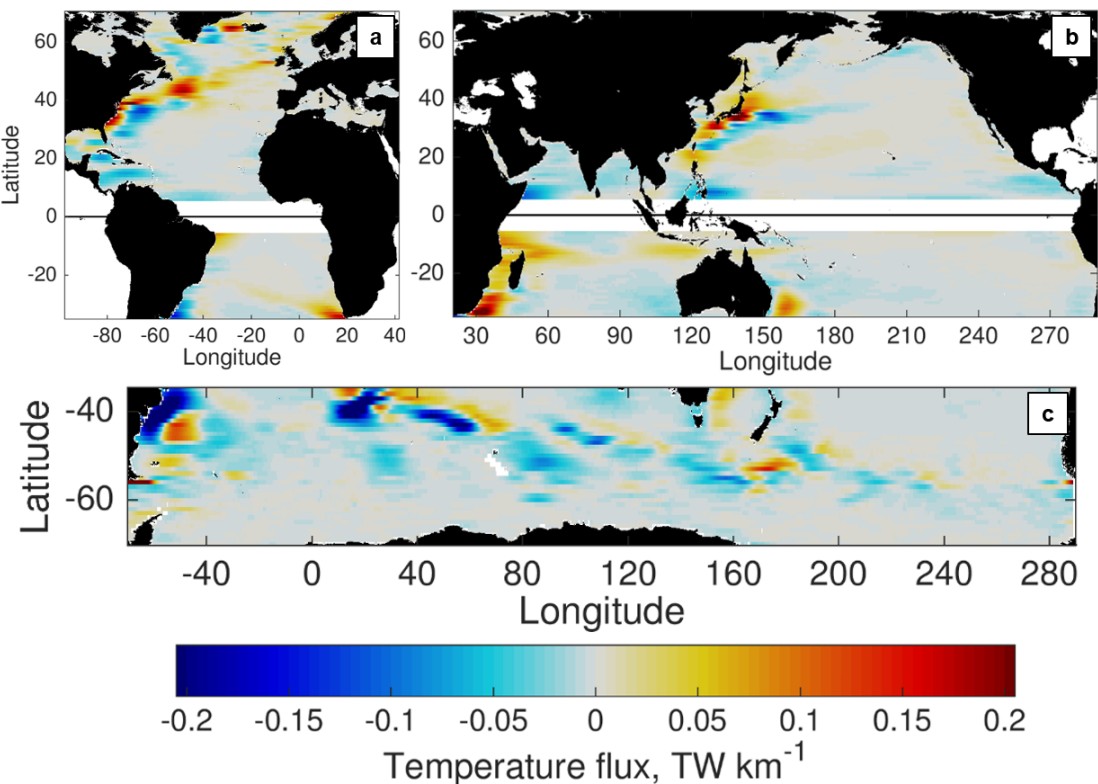

**Figure B2.** EKE-adjusted composite estimate of time mean zonally-smoothed meridional temperature flux associated with the mesoscale flow. Fluxes shown in the (a) Atlantic, (b) Indo-Pacific, and (c) Southern Ocean basins.

Mesoscale temperature fluxes based on these EKE-adjusted composites (Figure B2) imply some enhancements of fluxes relative to the unadjusted values in Figure 2 in the tropical and subtropical Indo-Pacific. These enhancements are evident
mostly in the subtropical eddy bands and at the Pacific eastern boundary near Central America. However, the overall patterns of the MTF (and in most areas the MTF magnitudes) are not affected by the adjustment. Furthermore, the MTF was zonally integrated in each basin to determine how the adjusted composites affect meridional HT (Figure B3). At nearly every latitude, the adjustments to meridional HT in the composites are very minor. The most substantive adjustments can be found near the meridional HT transport peaks in the Indo-Pacific tropics (north and south of the equator) and in the Southern Ocean near
40°S; the peaks near Indo-Pacific 13°S and Southern Ocean 40°S are enhanced by 0.1–0.15 PW (Figure B3b,c). Everywhere else, the adjustments are less than 0.1 PW. These results imply that, while the low time-mean EKE bias in the model may have

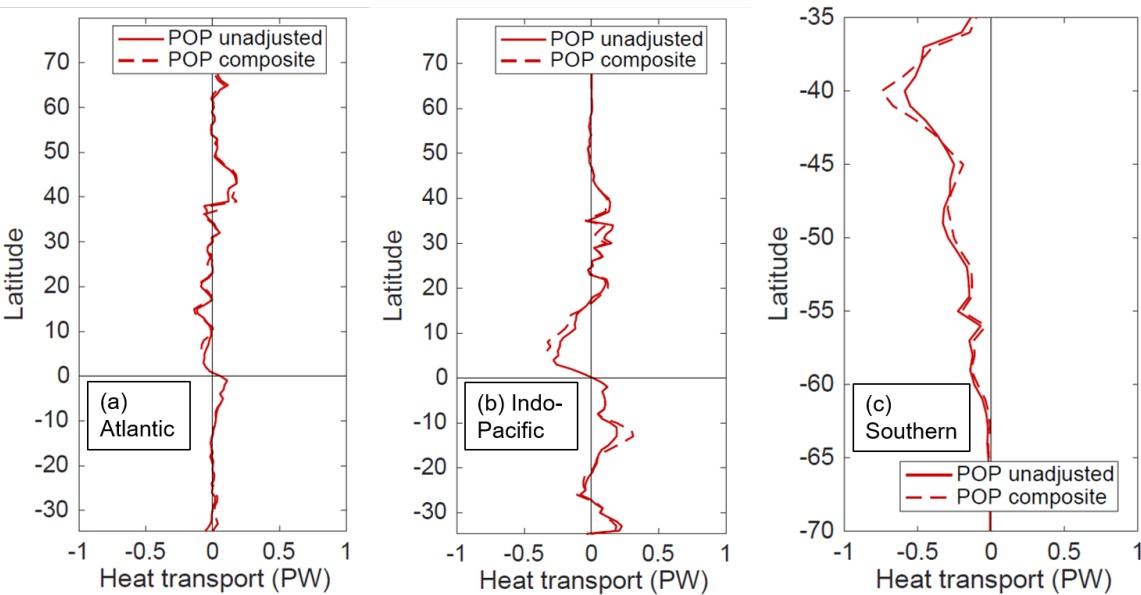

**Figure B3.** Mesoscale contribution to meridional HT in the (a) Atlantic, (b) Indo-Pacific, and (c) Southern ocean basins. The unadjusted values (corresponding to the red curves in Figure 7) and the values associated with the EKE-adjusted composites are shown at each latitude.

an impact on MTF values in some regions, the overall assessment of mesoscale contributions to meridional HT should not be greatly impacted by these EKE biases that occur mostly in interior ocean regions.

*Author contributions.* Primary author Andrew Delman wrote the code, carried out the analysis presented, and drafted the manuscript. Tong Lee supervised the project, providing input into the direction of the research and edits to the manuscript.

*Competing interests.* There are no competing interests present in the publication of this paper.

*Acknowledgements.* The research was carried out at the Jet Propulsion Laboratory, California Institute of Technology, under a contract with the National Aeronautics and Space Administration (80NM0018D004) with the support of NASA Physical Oceanography. The authors would like to acknowledge Benjamin Johnson who ran the POP model configuration and made the output available, as well as Frank Bryan at the National Center for Atmospheric Research who helped us obtain the output. We are also grateful to three anonymous reviewers whose feedback greatly improved this manuscript.

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
