# Peer review of "Global contributions of mesoscale dynamics to meridional heat transport"

_Ocean Science, 2021_

## Author Comment (AC1)

**Response to review comment #1:**

this paper is interesting and deals with an important subject

it is well written and well analyzed

my main concerns unfortunately pertain to the bases of the methodology

Thanks very much to the reviewer for their comments. We have described our revisions to the manuscript to address these comments below. Note: The responses to each of the comments are given in red text. Line numbers in the responses refer to the "tracked changes" version of the revised manuscript, so the line numbers are different in the version without tracked changes.

1) it is extremely rate that a high resolution numerical model has lower EKE level than AVISO (here 0.1 deg model for 0.25 deg altimetry)

this casts a serious doubt on the results of the paper if the level of mesoscale turbulence is not well represented (then the heat fluxes may have a major bias)

It is not uncommon for surface EKE to be underrepresented in models compared to satellite observations, even in "eddy-permitting models" such as  $1/10^{\circ}$  POP. As noted in the revised text (line numbers 106-107):

"However, the low bias in time-mean EKE is not unique to this eddy-permitting model (e.g., Volkov et al. 2008, 2010)."

This happens because even though the numerical model has a higher-resolution grid spacing than the observational data product, the model may not actually resolve more dynamics. We would only expect to see the model simulate phenomena which have wavelengths encompassing many grid points.

To address (in part) the concern about the low EKE bias in the model, we have constructed alternate estimates of time-mean temperature fluxes and heat transport (Appendix B) by averaging fluxes only during times when the zonally-smoothed EKE distribution is higher, such that the mean EKE during "composite" time periods closely resembles observed EKE (Figure B1). It is found that the change in meridional heat transport from this adjustment (Figure B3) is not that substantial, probably because most of the low bias in EKE occurs in quiescent parts of the ocean that are contributing relatively small mesoscale temperature fluxes.

We also note that many of our key conclusions are derived from regions where the model EKE is not seriously impacted by this bias (the Southern Ocean, most western boundary currents). Most of the fluxes associated with the stationary mesoscale circulation (Figure 3) occur in these areas where the model EKE is realistic; these fluxes are substantial, even though temporal definitions of the "eddy" flux exclude them. This is mentioned in the revised conclusions (Section 6):

"While POP simulates EKE well in the Southern Ocean and western boundary currents where MTF is substantial, the model has a low EKE bias in some subtropical and tropical regions (Figure 1). Hence the spatial decomposition method should be applied in models that simulate mesoscale activity and variability more accurately in these regions." (line numbers 569-571)

2) using lambda\_0 = 10 deg to separate large scale from mesoscale is in my opinion, really large (e.g. compared to the first internal radius of deformation or to the Rhines scale at say 15N) - at 15N-30N this leads to lambda0=800-1000km with lambda0/2=D the eddy diameters = 400-500 km which is really large

First of all, the  $10^{\circ}$  scale threshold we chose is meant to represent wavelength threshold, which corresponds to a 2.5° radius threshold. We agree that this is still a factor of two larger than the typical (dominant) eddy scales (e.g., Chelton et al. 2011). We chose this threshold intentionally to minimize leakage of large-eddy signals into large scales. In early uses of this methodology we tried a threshold that was closer to typical eddy scales, but some of the variability associated with larger eddies was leaked into large scale v and T profiles. It can be seen in the spectra of meridional velocity (Appendix A, Figure A1) that the mesoscale peak is fairly broad. A typical eddy radius of 120 km at 20° latitude (Chelton et al. 2011 figure 12) corresponds to a wavelength of 480 km or 4°-5° longitude; however the velocity spectra show that variance remains elevated from the mesoscale peak (at ~4° wavelength) to almost double this scale (panel e in the spectra plot). A separation scale of 10° longitude (and larger near the equator) therefore assures that larger eddies will still be classified as mesoscale, and be treated distinctly from gyre-scale motions and long planetary waves.

See also this quote added to Appendix A:

"The wavelength (not radius) lambda\_0 was deliberately chosen to be larger than the size of nearly all eddies, in order to ensure that their signals are retained in the mesoscale. For instance, an eddy radius of 1° longitude and diameter of 2° is typical at mid-latitudes (Chelton et al. 2011), corresponding to a wavelength of 4° longitude. If lambda\_0 at mid-latitudes is chosen to be 4° or 5° longitude, much of the signal of larger eddies will be included in the large-scale profiles along with basin gyres and long planetary waves. Hence a choice of lambda\_0 = 10° results in a cleaner separation of these phenomena." (line numbers 601-605)

therefore I advise to reconsider these two aspects

---

## Author Comment (AC2)

**Response to review comment #2:**

This paper decomposes the global heat transport into basins-scale, large-scale and mesoscale transports. Th main thing is to do so using a spatial filter, rather than the more common time-mean approach. The spatial filter results in different values for eddy heat transports than the time-mean method. However, little discussion is provided o why and how this benefits future work and what the differences really mean. Also, it is stated that this work help parameterization, but it is unclear how exactly. Overall, this paper could potentially be published, but it needs a clearer narrative, fewer figures, and improved use of symbols and names. Also, the importance of the results should be better explained and parts that are doubtful should be removed or require further argumentation.

Thanks very much to the reviewer for their comments. We have described our revisions to the manuscript to address these comments below. Note: The responses to each of the comments are given in red text. Line numbers in the responses refer to the "tracked changes" version of the revised manuscript, so the line numbers are different in the version without tracked changes.

Overall, I would recommend reducing this paper by a couple of figures and with that be more to the point about the most important results here. What is the main story and which figures do you need to prove this?

The key results of the paper, as clarified in the abstract, include:

- The mesoscale-induced divergence of heat from the subtropics towards the equator and the poles, caused by both stationary and transient mesoscale phenomena.
- The differences between spatial and temporal definitions of the eddy flux. The temporal eddy flux does not include stationary mesoscale recirculations, while temporal eddy flux includes long planetary wave contributions but mesoscale flux does not).
- Interannual/decadal variability in the mesoscale contributes to interannual/decadal variability of heat transport primarily in the North Atlantic, tropical Indo-Pacific, and the Southern Ocean (ACC).
- Eddy kinetic energy (EKE) is not a reliable proxy for mesoscale temperature fluxes in energetic high-EKE regions, though it is in some regions of low or moderate EKE.

Figures that do not relate directly to these key points have been removed. This includes Figures 3, 6, 13, and 15 in the previous manuscript. Figure 2 (spectra of meridional velocity at various latitudes) has been moved to Appendix A, in order to allow the manuscript to proceed more quickly to key results while retaining this information to justify the choice of threshold wavelength used. Figure 14 (Indo-Pacific time series) was streamlined to focus only on the situation at 4°N, where the mesoscale heat transport variability is aligned with ENSO. Some figures and their discussion in the text have also been reordered in order to highlight and discuss the key points above.

The concluding discussion in Section 6 has also been revised to clarify the ways in which these key results further the study of mesoscale fluxes and their representation in models:

"By implementing this diagnostic to separate scales in mesoscale-permitting and mesoscale-resolving simulations, it is possible to better understand how stationary and transient mesoscale phenomena induce fluxes across large-scale gradients. Future studies can focus on relating large-scale gradients (in one, two, and three dimensions) to mesoscale diffusivities, and therefore on improving parameterizations of mesoscale impacts from stationary recirculations as well as propagating eddies." (line numbers 563-567)

Rewrite section 2.2 to start with the most common method to define eddies as given in equation 10. From there introduce the spatial filter method. Then explain all the different types of eddies that are used in this paper (including eq 7,8,9) and more importantly, give them clear and distinct names. At this moment mesoscale is used in various different ways and definitions and it becomes very unclear.

Section 2.2 has been reorganized as suggested, starting with the definition of the temporal decomposition, then the discussion of the spatial decomposition that we use. Finally, the decomposition of the mesoscale temperature flux into stationary and time-varying components (eq. 7-9 in the earlier version) is discussed.

I keep finding the MTF confusing as in such contexts M often stand for Meridional and not Mesoscale. Meridional is also part of this study. It distracted me a few times.

Good point: since both meridional and mesoscale are used a lot in the paper it is tempting to use M to abbreviate both of them, but this may be confusing. In the revised version, mesoscale temperature flux is still abbreviated as MTF, but meridional heat transport is abbreviated as "meridional HT".

Specific (but still some major) comments

Eq 5&6: explain the difference.

The equations (now 6 and 7) are complementary to each other; equation (6) is a low-pass filter and equation (7) is a high-pass filter. The sum of the two resulting components is the original unfiltered data. This is now explained in the text:

"The sets of Fourier coefficients $V_L$ and $V_M$ resulting from the two filters sum to the original coefficients $V''$." (line numbers 159-160).

Figure 4 – the different axis-range for the y-axis is not helpful.

In this figure (Figure 7 in the revised manuscript) the y-axis range has been standardized for the three panels, ranging from -2 to +2.2 PW. The subsequent Figures 8-10 dealing with meridional heat transport have also had the y-axis ranges standardized.

L160 – This discussion is about the variability, which is as large as the transport itself. Please explain what that implies and what it says about using this filtering method.

This discussion relates to a figure (Figure 8 in the revised manuscript) showing the interannual/decadal variability of the spatial components of meridional HT at various latitudes. It is not clear whether the reviewer's comment refers to the variability of the mesoscale contribution or the total variability. In either case, that the temporal variability is as large as the transport itself does not say much about the filtering method, since the spatial decomposition is applied in the zonal dimension and not the time dimension. The large interannual variability in the Walker circulation (e.g., ENSO) likely accounts for much of the large total (mostly overturning) variability at low latitudes, while the ID variability of the mesoscale contribution are likely to be the result of interannual changes in eddy and TIW activity, as stated in the text:

"The largest peaks in ID mesoscale meridional HT variability occur at 43°-40°S, 3°-4°N, and 32°N, with the latter two peaks explained mostly by contributions from the Indo-Pacific (Figure 8c). These latitudes correspond to the high EKE regions associated with the Agulhas Return Current, Pacific TIWs, and Kuroshio respectively, implying that the very active mesoscale dynamics and interannual variability in these areas result in large MTF variability on ID timescales as well." (line numbers 389-395)

L179 – Why these numbers for s and k0, and what do they mean? Are results sensitive to these choices?

We have provided clarifications in the revised text after eq. (6) and (7):

"The forms of the low-pass (6) and high-pass (7) filters are symmetric and contain a steepness factor s that affects the rate of signal roll-off near the cutoff wavenumber; a higher value approaches a boxcar filter with associated "ringing" effects, while a lower value avoids ringing but at the expense of cutoff precision. In filtering for large-scale and mesoscale v and T, a value of s = 5 was selected as an optimal balance of roll-off between the physical coordinate and spectral wavenumber domains." (line numbers 154-159)

And in the discussion of the smoothing filter: "…for this smoothing the threshold wavenumber $k_0$ is chosen to be half the threshold wavenumber (twice the threshold wavelength) for the large-scale/mesoscale separation at that latitude. This larger value of the threshold wavelength is used to remove more of the rotational fluxes that occur at the mesoscale. In the smoothing filter the steepness factor s is also set to 2 (rather than 5), because the smoothing effect is more important here than the precision of the wavelength cutoff." (line numbers 264-268)

L182-184 – Are you sure about this statement? If so, what is your evidence for this?

Coarse-resolution models only simulate the large scale explicitly, not the mesoscale. So in order to improve their accuracy, we need to parameterize the effects of the non-resolved dynamics (mesoscale and smaller scales) on the resolved-scale structure.

L188 – why are they opposed?

The figure being discussed (large-scale temperature fluxes) has been removed from the revised manuscript in order to focus on the mesoscale temperature fluxes.

L198 – Why important in eastern boundaries?

Again, the figure being discussed has been removed from the revised manuscript.

L206 – Are you sure? Could it not also be the other way around? How does this work, what is the mechanism?

This discussion was a comparison of maps of the large-scale temperature flux vs. the mesoscale temperature flux. Since the maps of the large-scale temperature flux are no longer in the revised paper, this discussion has been removed from the text.

L220 – Over how long do you average? I guess you come back to that later. Perhaps refer here.

As mentioned towards the end of the revised Section 2.2: "In this analysis the time averages are applied over the full 32 years of model output used." (line number 234)

Fig 8 – Provide Title this is only Indo-Pacific. Perhaps indicate major continents or so in axis. Are all these figures needed to tell you story?

This figure (Figure 5 in the revised manuscript) has been pared down to focus on the cumulative zonally-integrated temperature fluxes that sum up to the heat transport in the basin. The figure caption mentions that this is for the Indo-Pacific basin, and some major features (e.g., currents, marginal seas) have been annotated on the figure panels.

L253 – I find "time-varying" a bad choice, as they are both time varying.

This flux resulting from the temporal decomposition only is now referred to as "all time-varying" to distinguish it from the "high frequency" flux, as well as from the mesoscale time-varying flux $MTF_{vary}$.

Figure 10 – "mesoscale components" by now I'm so confused as to which components we are dealing with. Temporal, spatial, variability, mean, etc. etc.

Section 2.2 has been reorganized as suggested by the reviewer, with clearer terminology introduced in that section. Hopefully this resolves the confusion.

L261 – Very loose statement about Kelvin and Rossby waves. Explain the physics and provide references.

Text has been added explaining that long Kelvin and Rossby waves "are time-varying with frequencies ranging from intraseasonal to interannual, but with spatial wavelengths spanning an entire ocean basin (e.g., Boulanger and Menkes 1999; McPhaden and Yu 1999)". (line numbers 412-414)

L262 – This is also a loose stamen. It may be clear from the results, but it is not explained why, and why one is preferred over the other, and for what situation such preference might be true or false. In other words, it is not yet shows that spatial filter is better than temporal, nor explained exactly way the reason is behind al the differences. Only shown it is different.

Is the reviewer referring to the statement that was in lines 261-262 itself, or asking a more general question about why the spatial decomposition would be considered preferable to the temporal decomposition?  The statement that was in lines 261-262 has been refined slightly to specify that long planetary waves (especially in the tropics) have very different characteristics from mesoscale eddies.

As for the latter point, the motivation for a spatial decomposition is stated in Section 1:

"…these temporal decomposition methods may conflate the contributions of large-scale and mesoscale circulations, as gyres and planetary waves have temporal covariances between v and T. Moreover, the effects of stationary mesoscale features (e.g., recirculation gyres) are not included in the temporal eddy meridional HT contributions.  Since spatial resolution prevents climate models from explicitly simulating mesoscale ocean dynamics, accurate representations of the ocean depend on parameterizations (e.g., Gent and McWilliams, 1990; Eden and Greatbatch, 2008; Marshall et al., 2012) that must take into account stationary as well as transient mesoscale fluxes." (line numbers ?)

Figure 12 – Which mesoscale flow is shown and how much it is smoothed?

The zonal smoothing filter is the same one used in earlier maps of the time-mean mesoscale temperature flux; this has been clarified in the figure caption (Figure 11 in the revised manuscript): "zonally smoothed using the same smoothing filter as in Figures 2-4".

L280 - I'm not convinced about the need of the paragraph starting at L280 and the associated Figure 13. Isn't the variation only 10-20% of the total at the peaks only. Is this important? Perhaps it is, but it is not clear here.

This discussion has been removed from the manuscript along with the figure that it is associated with.

Figure 14 – Clarify in title that OT is removed in lower panels.

This has been clarified in the figure caption (Figure 13 in the revised manuscript): "(b) Same as (a), but with the overturning component of meridional HT variability removed".

L317 – How substantial is the variability? To what are we comparing this and how much is that percentagewise or what is the correlation factor?

The $R_{MTF}$ (as mapped in Figure 12 in the revised manuscript) reflects both the amplitude of the variability and its correlation with the total heat transport at that latitude, as indicated in eq. (11). In addition to the discussion of RMTF, the text also mentions two regions where the correlation itself is particularly high: "…there are two regions where the local MTF and basin-integrated total meridional HT are also highly positively correlated with C(loc,tot) > 0.5; the Pacific TIW region north of the equator, and the lee of the Kerguelen Plateau.  The implication is that mesoscale processes in these two regions have a direct impact on the meridional HT at their respective latitudes, without much compensation from the overturning or large-scale contributions." (line numbers 504-507)

Section 5.2

I'm not convinced that this section is correct. It is based on a very big IF in L340. Check papers on changing Rossby radius of deformation and temperature on short and large scales. They both vary strongly with y. I'm not convinced this part is very meaningful, without more evidence that this could work.

The reviewer is correct about the "very big IF", as mixing length and meridional temperature gradient may indeed vary greatly.  Yet Figure 14 in the revised manuscript shows that, in spite of these caveats, EKE explains 50% or more of the interannual/decadal heat transport variance in some moderate eddying regions (subtropical eddy bands, southern excursions of the ACC). Hence the results in Section 5.2 indicate where the assumption that mixing length and temperature gradients are fairly consistent in time is valid, and where the assumption is not valid (energetic western boundary currents).

A sentence has been added to the text to emphasize this: "Figure 14 does however provide an indication of areas of the ocean where surface EKE might be a decent proxy for the MTF, in contrast with the more energetic regions where mixing length and gradient variability can interfere with the EKE-MTF relationship." (line numbers 540-542)

---

## Author Comment (AC3)

**Response to review comment #3:**

The study presents findings about the contribution of mesoscale dynamics to the global meridional heat transport in the Parallel Ocean Program 2 ocean model. Contrary to most previous papers that use filters in the time domain to isolate the mesoscale signal, the authors use a spatial filter to separate velocity and temperature fields into overturning, large scale and mesoscale components. They identify global patterns and key regions with significant eddy heat fluxes and address the variability of eddy heat fluxes on interannual to decadal time scales.

Overall, I think the results and questions raised by the paper are very relevant for the community. However, some improvements are necessary before I can recommend the paper for publication.

Thanks very much to the reviewer for their comments. We have described our revisions to the manuscript to address these comments below. Note: The responses to each of the comments are given in red text. Line numbers in the responses refer to the "tracked changes" version of the revised manuscript, so the line numbers are different in the version without tracked changes.

**General comments:**

- While I think the spatial separation into OT, M and L component is definitely interesting and should be further studied, I am not yet convinced that the method (at this stage) is "better" or "worse" than separations in the time domain. For example, section 3.3 shows that the largest signal in regions of high MHT stems from stationary transport in the western boundary currents (which would not be considered mesoscale when separating the scales in time domain). In my eyes it is at least questionable whether the time mean heat flux of a boundary current (even if it is only a few hundred km wide) should be considered an "eddy signal". Section 4 shows the differences between the methods (which is definitely very interesting), but there are no reasons provided for the observed differences and no evaluation of which method delivers the more realistic results.

Which method delivers more "realistic" results depends on the question that is being investigated. In this study, we focus on quantifying the impact of mesoscale dynamics, rather than exclusively eddies, or the time-varying contribution. There are two main motivations for this focus:

(1) As discussed in the introduction (line numbers 58-61), climate models often do not resolve the oceanic mesoscale, and need to parameterize their impacts. Spatial (not temporal) scale is generally the limiting factor that prevents these models from simulating eddies and other mesoscale phenomena (this is why the spatial resolution of models is mentioned much more often than their timestep). If the models are limited by spatial scale, then it is not only the effects of time-varying eddies that need to be parameterized, but also the effects of stationary mesoscale recirculations. Effective diagnostics are needed to quantify the fluxes associated specifically with mesoscale spatial scales. These diagnostics can be applied to models that *can* resolve these scales, to constrain the parameterized fluxes in models that *can not* resolve the mesoscale. This is discussed in Section 6: "By implementing this diagnostic to separate scales in mesoscale-permitting

and mesoscale-resolving simulations, it is possible to better understand how stationary and transient mesoscale phenomena induce fluxes across large-scale gradients. Future studies can focus on relating large-scale gradients (in one, two, and three dimensions) to mesoscale diffusivities, and therefore on improving parameterizations of mesoscale impacts from stationary recirculations as well as propagating eddies." (line numbers 563-567)

(2) Despite abundant work on parameterizations of mesoscale-induced lateral mixing (e.g., Gent and McWilliams 1990; Eden and Greatbatch 2008; Marshall et al. 2012), an intuitive understanding of how energy at mesoscales relates to tracer fluxes is still needed. Stationary mesoscale phenomena (such as boundary current recirculations) are quite different structurally from transient eddies, but both are forms of nonlinear mesoscale turbulence. As discussed in the introduction:

"Unlike the wind-forced response associated with larger-scale gyres and planetary waves, these mesoscale phenomena are generated and sustained by nonlinear mechanisms such as baroclinic and barotropic instability (e.g., Eady, 1949; Charney and Stern, 1962) and nonlinear momentum advection (e.g., Greatbatch et al., 2010)." (line numbers 27-30)

By developing diagnostics to isolate the fluxes associated with both stationary and transient mesoscale flows, we can better understand why an increase in mesoscale energy may not increase the mesoscale flux (as suggested in Section 5.2), or even why upgradient fluxes might occur.

• Using a fixed value of 10° longitude to separate between the large scale and mesoscale components of temperature and velocity fields poleward of 20° seems quite large. It should be appropriate in mid-latitudes, but the model domain extends far into the subpolar regions where eddy scales are much smaller. How sensitive are the results to this choice?

To be clear, the scale we are referring to is wavelength; the wavelength of an eddy field is approximately 4 times the radius of a given eddy. Hence, a typical eddy radius of 120 km at 20° latitude (Chelton et al. 2011 figure 12) corresponds to a wavelength of 480 km or 4°-5° longitude.

However, we intentionally chose a threshold for separating the large scale and mesoscale that is much larger than typical eddy scales (e.g., Chelton et al. 2011). In early uses of this methodology we tried a threshold that was closer to typical eddy scales, but some of the variability associated with larger eddies was leaked into large scale v and T profiles. It can be seen in the spectra of meridional velocity (Fig. 2 in old manuscript, moved to supplement) that the mesoscale peak is fairly broad. The velocity spectra show that variance remains elevated from the mesoscale peak (at ~4° wavelength) to almost double this scale (panel e in the spectra plot). A separation scale of 10° longitude (and larger near the equator) therefore assures that larger eddies will still be classified as mesoscale, and be treated distinctly from gyre-scale motions and long planetary waves. Moving poleward from 20° to 60°-65° latitude, the spectra plots also show that the scales of the

mesoscale peak and transition remain fairly consistent in degrees longitude. This is because the reduction in eddy length scales at mid-to-high latitudes occurs at a similar rate to the reduction in spacing between lines of longitude. So the eddy scales are much smaller at 60° than 20° latitude, but not when measured by degrees longitude.

See also this quote added to Appendix A:

"The wavelength (not radius) lambda_0 was deliberately chosen to be larger than the size of nearly all eddies, in order to ensure that their signals are retained in the mesoscale. For instance, an eddy radius of 1° longitude and diameter of 2° is typical at mid-latitudes (Chelton et al. 2011), corresponding to a wavelength of 4° longitude. If lambda_0 at mid-latitudes is 4° or 5° longitude, much of the signal of larger eddies will be included in the large-scale profiles along with basin gyres and long planetary waves. Hence a choice of lambda_0 = 10° results in a cleaner separation of these phenomena." (line numbers 601-605)

- The paper is quite long, and I find some parts and especially some figures more distracting than helpful. The story needs to be streamlined to focus more closely on the main results and their relevance. For example:

  - section 2.3 includes 6 plots, but it did not convince me that 10° longitude is indeed a good scale for the separation of large- and mesoscale

  In order to streamline and focus the paper more on the key results, this figure has been moved out of the main body of the paper to Appendix A.

  - section 3.1 only shows that the model gets the general ocean circulation right, that should be a given for an ocean model and is not relevant for the results of this study

  This figure is being removed for the sake of brevity.

  - section 5.1 has 14 individual time series plots that basically only show when total MHT and mesoscale MHT are in phase. They are also really hard to read on a printout

  This figure has been pared down significantly, to show only the 4°N transect in the Indo-Pacific basin, and the influence of ENSO on the mesoscale contribution to meridional heat transport.

**Detailed comment list:**

Line 13: This assumes that dT/dy and mixing length are constant (compare line 340), which is a strong assumption. Maybe rephrase to something like "surface EKE alone is not a good proxy..."

Yes, this change has been implemented. (lines 16-17)

Line 42: I would suggest adding Sun et al. (2019) as a reference here. They did a global analysis regarding the importance of the different transport mechanisms (swirl vs advection)

This reference has been included. (line 46)

Line 51: If the recirculation is 100% perfectly stationary, it would be "invisible" in a time filtered mesoscale component, but it would show in the time-mean component. And if there are small fluctuations in v and T, there can be covariance and thus an "eddy" heat flux.

Yes, the effects of "wobbles" in boundary currents and associated recirculations would be included in a time-varying temperature flux. But even the fluxes associated with perfectly stationary mesoscale recirculations will still need to be parameterized in coarse-resolution models that do not resolve them.

Line 65: A visual comparison is not a validation

The word "validation" has been changed to "assessment" in this section title.

Line 68: How many depth levels?

The text is augmented to say "The model simulation has 62 depth levels with 10 meter vertical spacing in the upper 160 meters". (line numbers 87-88)

Line 70: If I understand the tripolar grid correctly, "progressively finer" is quite misleading here. The resolution is still 0.1°, but due to the convergence of the grid towards the two poles, the distances (in km) between two grid cells become smaller. It might be helpful to compare the grid resolution in km to the local Rossby radius to check where the model resolution is fine enough to resolve eddies (see e.g,. Wekerle et al. (2017), their Fig. 2).

The text has been revised to give a sense of the grid spacing at higher latitudes: "…progressively finer grid spacing approaching the two north poles. In physical distance, the grid spacing is approximately 11 km grid spacing near the equator, 5.5 km at 60°S and 5-7.5 km grid spacing at 60°N. Given this spacing relative to the baroclinic deformation radius (e.g., Hallberg 2013; Wekerle et al. 2017), it is expected that the model should at least permit mesoscale instability development equatorward of 50°-60° latitude." (line numbers 83-87)

How is the tripolar grid considered when calculating the temperature fluxes? Is the velocity field rotated to north or are the deviations between perpendicular to the grid line and north so small that it does not matter?

The following text has been added to Section 2.2 on the methodology: "In this analysis, transects are extracted from the model output along tracks corresponding to the nearest grid faces to integer lines of latitude. In the non-Mercator portions of the grid in the Northern Hemisphere, this results in a zig-zag of the transect through the model grid, but in this way the net volume

transport through each transect is conserved." (line numbers 145-147)

I also think this section should focus more on the model with respect to its "eddy capability", how well and where the model represents mesoscale processes (comparison with EKE is a good start). With 1/10° horizontal resolution the model can't be considered eddy resolving everywhere, so there will still be sub grid-scale processes that aren't resolved by the model. How does that affect the interpretation of the results? How are they parameterized on the sub grid-scale? What is the eddy diffusivity and viscosity for lateral mixing and diffusion in the model simulation?

The following text has been added to Section 2.1:

"One possible explanation is that the biharmonic viscosity $\nu_0 = -2.7 \times 10^{10}$ m$^4$ s$^{-1}$ and diffusivity $\kappa_0 = -3 \times 10^9$ m$^4$ s$^{-1}$ values used in this model simulation are well tuned for energetic regions such as western boundary currents, but may suppress too much mesoscale activity in the less energetic ocean interiors." (line numbers 104-106)

Composite estimates of MTF and heat transport have also been computed using a subset of times in the model output. In regions where POP underestimates EKE, higher EKE times are composited so that the EKE time average during the composite periods is closer to the values from altimetry. These composites and an explanation of them are discussed in Appendix B.

Fig. 1: I am a bit surprised by how much lower the model EKE is when compared to CMEMS altimetry. Why is the model velocity field filtered (wavelengths < 0.5°)? Would that explain the low EKE? What filter is used here? The same 1D zonal filter as described in 2.2 or a different one?

The filter that is used is a two-dimensional (zonal and meridional) application of the low-pass filter in eq. (6), and this is indicated in the figure caption.

The following text has been added to the discussion of Figure 1: "The underestimation of EKE in POP persists even if a different filter wavelength is applied to both model and altimetry datasets (e.g. 1°), and indicates a consistent low bias of mesoscale activity in lower-energy parts of the ocean interior." (line numbers 102-104)

Line 90: The separation happens before integration

This is clarified in the text: "In the spatial decomposition used in this study, the meridional temperature flux at each depth level is decomposed into zonal mean and zonal deviation components…The overturning component, when multiplied by (rho)(c_rho) and vertically integrated, gives the portion of meridional HT associated with basin-wide vertical gradients in meridional flow and temperature." (line numbers 125-131)

Line 92: What about the z-direction? Are all values depth integrated or is this separated at every depth level and then integrated over z? Is the heat transport integrated over the full water column or limited to the surface?

The decomposition occurs at each depth level, before vertical integration; this is clarified in the text (quote above; line numbers 125-126).

Line 110-117: What is the benefit of using this particular filter method compared to other filters? E.g. Zhao et al. (2018) use a Butterworth window for their filter?

The results are not particularly sensitive to the exact filter used, provided the roll-off of the filter is not too steep in either physical or spectral (wavenumber) space, so that any ringing is minimized. The form of the filter in equations (5)-(6) is chosen as the low-pass and high-pass forms of the filter are symmetric to each other (this can be seen by comparing eq. 5 & 6), and there is one easily-varied parameter s that can be tested to find optimal roll-off at the separation scale. In this case we settled on a value of s = 5. This is mentioned in the revised text: "The forms of the low-pass (6) and high-pass (7) filters are symmetric and contain a steepness factor s that affects the rate of signal roll-off near the cutoff wavenumber…In filtering for large-scale and mesoscale v and T, a value of s = 5 was selected as an optimal balance of roll-off between the physical coordinate and spectral wavenumber domains." (line numbers 154-159)

Line 118: What about Zhao et al. (2018)? If this is what inspired using the spatial filter, maybe put this in the introduction or the beginning of section 2.2

The reference to Zhao et al. (2018) has been moved to before the spatial decomposition is introduced:

"This method was used by Delman and Lee (2020), following a similar application of spatial filters by Zhao et al. (2018), but with the additional separation of the overturning contribution and corrections to the filtered v and T profiles near lateral boundaries." (line numbers 119-121)

Line 119: Despite checking Delman & Lee (2020), I am not quite sure how the filter handles the basin boundaries. From Delman & Lee (2020): "Meridional velocity outside the basin boundaries (to a distance of 1/k0 from the outermost boundaries) and within interior land areas is set to zero.", "a buffer is also included at the western and eastern boundaries", "in order to conserve the large-scale structure of zonally integrated v, this non-zero vL needs to be redistributed over nearby water areas in the transect.". I must admit don't understand what has been done here exactly and I am not sure if all three steps are there to correct errors at the boundaries. However, getting the signal at the western boundary currents right is incredibly important since they are so dominant but confined within a small distance to the coast. I would therefore suggest clarifying this here.

The quotes the reviewer included from Delman and Lee (2020) refer to three steps that we have taken at the boundaries to improve both transect-integrated conservation of volume within each component, as well as local representation of the large-scale and mesoscale separation. These steps, plus the channel correction also described in Delman and Lee (2020), are applied either before or after the filters. Text has been added to the revision to briefly describe these corrections:

"To better preserve zero net volume/mass flux in the basin-integrated $v_L$ and $v_M$ components, our method incorporates boundary and channel corrections (described in more detail in Delman and Lee, 2020). These corrections also aim to improve local representation of the large-scale/mesoscale separation near boundaries and in narrow channels. Before filtering, (1) v profiles over land areas are set to zero, and (2) a buffer is applied to temperature profiles over land areas near boundaries to avoid sharp swings in $T_L$. After filtering, (3) non-zero $v_L$ and $v_M$ that leaked onto land areas is returned to water areas, and (4) within channels bounded by bathymetry that are narrower than $\lambda_0/4$, the meridional velocity profiles are set to $v_L = v$ and $v_M = 0$. Steps (1), (3), and (4) are taken in order to improve conservation of volume along the transect, while steps (2) and (3) improve the local representation of the large-scale and mesoscale separation." (line numbers 166-186)

Line 120-124: Is lambda/4 a good choice here? How do you separate the signal in a channel narrower than lambda into scales larger and smaller than lambda?

Note the text added to the end of Section 2.2:

"Regarding the rationale for step (4), at most latitudes (Figure A1), mesoscale activity peaks near wavelength $\lambda_0/2$, so $\lambda_0/4$ is approximately the diameter of a typical mesoscale eddy. Channels narrower than $\lambda_0/4$ are therefore too narrow to support typical mesoscale instabilities, but transport in these channels can contribute substantially to the large-scale circulation (e.g., Indonesian Throughflow, Gulf Stream in the Florida Strait). Hence all of the transport in these narrow channels has been assigned to the large-scale component." (line numbers 186-202)

Line 131: As mentioned above (general comment 2), 10° longitude for the separation between large- and mesoscale seems quite large. For example, 10° longitude at 50°N this is more than 700km. How does that compare to eddy diameters at this latitude? How sensitive are the results to this choice?

Eddy diameters at most latitudes are typically on the order of $\lambda_0/4$ as explained in the quote above. The results are not very sensitive to slight variations in $\lambda_0$ on either side of 10°, but if $\lambda_0$ was chosen to be 5° longitude instead (and this was attempted), a significant portion of the eddy v and T profiles would be leaked into the large-scale component. See the quote added to the text below in Appendix A, and also the response to general comment 2.

"The wavelength (not radius) $\lambda_0$ was deliberately chosen to be larger than the size of nearly all eddies, in order to ensure that their signals are retained in the mesoscale. For instance, an eddy radius of 1° longitude and diameter of 2° is typical at mid-latitudes (Chelton et al. 2011), corresponding to a wavelength of 4° longitude. If $\lambda_0$ at mid-latitudes is 4° or 5° longitude, much of the signal of larger eddies will be included in the large-scale profiles along with basin gyres and long planetary waves. Hence a choice of $\lambda_0 = 10°$ results in a cleaner separation of these phenomena." (line numbers 601-605)

What about filtering in the meridional direction? The final section mentions the rotational and

divergent eddy fluxes, but does it also affect the meridional coherence of the eddy heat transport if the underlying fields are only filtered in the zonal dimension? (I don't know the answer to this question.)

The filters used to separate the large-scale from mesoscale v and T are only applied in the zonal direction. Despite this fact, there is little evidence for significant abrupt breaks in the meridional coherence of MTF or the mesoscale contribution to HT. While applying a meridional filter might have an impact on the results in a few areas, it was decided for computational ease to only apply the filters zonally.

Fig. 2: How is the "mesoscale transition computed using the logarithmically smoothed spectral density" defined? What is the smoothing applied to the spectra? Why are 5 plots for the Indo-Pacific shown and none for the Atlantic? Since the same filter is applied globally, wouldn't it be enough to show one plot per latitude, or maybe even just one k/lat/spectrum-plot, where color represents the spectral density?

The spectral plots (former Figure 2, now Figure A1) have been moved to a supplement, in order to avoid distraction from the paper's focus on the mesoscale contribution results. The spectra have cleaner results when applied to each basin individually rather than along one transect globally, avoiding contamination from abrupt jumps that may occur at the boundaries. The idea of a k/lat/spectrum plot is interesting, but the change in spectral slope from the transition scale to the mesoscale bump/peak is more easily visualized when the spectra are plotted as lines.

Section 3.1 I think this whole section is unnecessary (see general comment 4). I would assume that an ocean model has gyres.

This section has been removed in the revised manuscript.

Line 147: I find the title "spatial components" of meridional HT a bit odd (because my mind automatically goes to spatial patterns and maps), but I don't have a better idea

The title has been changed to "Spatial decomposition of meridional HT". We agree that the term "spatial" could be more precise, but have not thought of a preferable alternative. "Zonal decomposition of meridional heat transport" is one possibility; more precise, but also potentially confusing with the juxtaposition of zonal and meridional.

Section 3.2 I would suggest restructuring this section starting with the maps (pointwise temperature flux) and then looking at the integrated values (basin wide heat transport). But that might just be personal taste

In the revised manuscript, the figures have been reordered as suggested.

Line 150: Is this related to eddies in the Agulhas Retroflection?

The maximum magnitude mesoscale contribution at 40°S is due not so much to the retroflection itself, but to eddies that form in the Agulhas Return Current and in the Brazil-Malvinas

Confluence region. As stated in the revised text: "The largest magnitude mesoscale contribution to meridional HT at any latitude is at 40°S, with a poleward heat transport (-0.6 PW) powered by strong MTFs in the Brazil-Malvinas Confluence and Agulhas Return Current regions (Figure 2)." (line numbers 367-370)

Fig. 4/5: Both figures have different y-ranges for the different ocean basins, which is a bit misleading. Where does the residual term come from? How is the transport integrated in the southern region? From South America to South America? And in the ACC?

The axis limits of these figures (Figures 7 and 8 in the revised manuscript) have been aligned so that they are the same in all panels. The residual term indicates the effect of high-frequency (timescales <5 days) co-variations in v and T that can not be spatially decomposed; this is described in the caption of Figure 7 in the revised text. In the southern region we integrate from 70°W across all longitudes to 70°W, though the choice of starting and ending point do not matter as long as they are the same.

Fig. 4: I am not so familiar with the Indian Ocean, but is it standard practice to ignore Australia and the Indonesian islands and just integrate over everything?

The Indian and Pacific basins are sometimes integrated together (e.g., Volkov et al. 2008) because HT is only physically meaningful when integrated across a transect through which the volume transport is negligible. This is true of the Atlantic, where the only non-zero volume transport in basin integrals is the Bering Strait overflow (~1 Sv). However, because of the Indonesian Throughflow (~15 Sv), the southern Indian and Pacific basins both have large volume transports, and integrating them together largely eliminates this problem.

Fig. 5: Why show the ID standard deviation here and not in the section titled "mesoscale interannual/decadal variability"?

We understand that the figure could also be grouped with the interannual/decadal variability maps, but thought that the manuscript flows better if the basin-integrated meridional HT plots are shown in sequence.

Line 179: How are these filter parameters chosen? Are the results sensitive to the choice?

For this smoothing filter, a threshold wavelength that is larger than the large-scale/mesoscale separation, in order to remove most of the rotational fluxes associated with the mesoscale. The value of s = 2 has a more gradual wavelength roll-off than the s = 5 used to separate the large-scale and mesoscale, since the objective of the smoothing is not to separate distinct spatial scales but to show the spatially-smoothed regional behavior of the MTF. As stated in the revised text: "In the smoothing filter the steepness factor s is also set to 2 (rather than 5), because the smoothing effect is more important here than the precision of the wavelength cutoff." (line numbers 267-268)

Fig 6: The colors seem to be quite oversaturated. Are the limits of the colorbar appropriate for the data?

This figure has been removed from the revised manuscript.

I am very surprised by the strong northward flux in the western Labrador Sea (Fig 6a). You mention that the "picture becomes more complex", but is there an explanation for this pattern?

This figure has been removed from the revised manuscript.

Line 225: "intensified boundary current jet and flanking recirculation just to the east that is shown in vM" and later in line 228: "stationary MTF contributions are associated with western boundaries"; Wouldn't that suggest that this is not really a mesoscale signal then? This relates to my general comment (1)

We make the case in this paper that jets and recirculations *are* mesoscale signals if their structures occupy similar spatial scales as mesoscale eddies, and their dynamics will not be fully resolved by coarse-resolution models. This reasoning has been described more fully in the revised text in Section 1, and is also explained in the response to your general comment 1.

Fig. 8: This figure seems to combine two thoughts and looks incredibly crowded to me. On the one hand, there is the separation into large- and mesoscale and on the other hand the separation into stationary and time-varying transport, leading to 12 subplots. Are the left and middle columns really necessary?

We agree with the reviewer that the figure was crowded, and the left and middle columns have been removed.

Why is the 95m depth level shown? Anything special about that layer?

The part of the figure that this comment refers to has been removed.

Fig. 9: This is basically just a zoomed in version of the middle column of Fig. 8. (at interesting locations). So maybe skip the middle of Fig. 8 and just show this? Why is the 95m depth level shown?

The left and middle columns of what was Figure 8 (Figure 5 in the revised text) have been removed.

Line 245: There are also methods looking at deviations from the seasonal cycle

Yes, fluxes can be decomposed into seasonal and non-seasonal components as well (e.g., to understand the contributions of advection by non-seasonal v and T to the seasonal cycle). This broadly fits under the umbrella of a temporal decomposition since v and T are separated using a time averaging operator. A brief mention of using the seasonal average has been incorporated into the discussion in Section 2.2. (line number 117)

Line 256: Mesoscale dynamics indeed have higher frequencies in lower latitudes leading to shorter eddy time scales there. You can also just look at eddy speeds along their trajectories for that.

Yes, the faster eddy propagation speeds have been mentioned in the text. (line number 409)

Line 261: Yes, that is indeed a problem. However, as shown in Fig. 8, the separation in the spatial domain makes the time-mean heat transport of the western boundary current an "eddy" signal, even though western boundary currents have very different dynamics from mesoscale eddies. See general comment 1.

This is a good point; the dynamics of planetary waves, western boundary currents (WBCs), and mesoscale eddies are all distinct, so why lump WBCs together with mesoscale eddies? One outcome of this paper is to identify characteristics of WBCs that have mesoscale spatial structure, the contributions of which may not be well represented in coarse-resolution climate models (see response to general comment 1).

The text has been revised to clarify the type of distinction we are referring to: "Hence the temporal eddy fluxes conflate the contributions of large-scale planetary wave activity and mesoscale eddy activity. In contrast, the mesoscale component isolates the contributions to meridional HT from fluxes that are not expected to be well represented in coarse-resolution climate models." (line numbers 414-417)

Section 5.1 This section is again very long and consists of different ideas and concepts (first a map of STD, then time series, then linear regression), including 5 different figures and 18 subplots. What is the main message of this subsection? Can it be conveyed in simpler ways?

Most of the time series plots in this section have been removed to streamline the section and the manuscript as a whole. In the revised manuscript, the standard deviation plot is followed by the linear regression plot to illustrate where MTF variability actually contributes to overall meridional HT; then time series in the tropical Indo-Pacific are included to indicate the influence of ENSO on MTF in this region.

Line 278: "This indicates that mesoscale fluxes are not be particularly efficient at moving heat meridionally in subtropical eddy bands, at least in the POP simulation." This could also be related to the small meridional temperature gradients in the subtropics

The small meridional temperature gradients are likely part of the story, and may also explain why the meridional HT remains largely unchanged even when only higher-EKE times are averaged (Figure B3). The typo in the above line has also been corrected.

Fig 13/14/15: Maybe show only the lower panels and focus on one section per basin? What is the reason for showing all these time series other than "sometimes mesoscale MHT and total MHT are in phase" The figures are super small and on a regular printout I can't really say anything about them. Zooming into the pdf, I can say that I disagree with some of the red dashed lines (e.g., the last two events in Fig 15 b).

Most of the panels in these figures have been removed; only part of the middle figure has been retained to show the connection between ID variability in the tropical Indo-Pacific and ENSO.

Line 303-329: I am not quite sure if this part is needed. If it stays in the paper, it should probably have its own section because it is quite different from the analysis of the transport time series before.

Most of the time series have been removed, so that this section now focuses more specifically on the mesoscale variability and its regional distribution.  The ID standard deviation map (Figure 11) is now followed by the regression map (Figure 12), and then the time series at one latitude (4°N) showing the influence of ENSO.

Line 320-323: Does this mean that the separation into large- and mesoscale is off in these regions?

Not necessarily; a change in the large-scale structure may precondition the background gradients on which the mesoscale fluxes occur.  For example, if a large-scale current is more zonally-oriented than usual, the large-scale temperature flux in the meridional direction will be lower than usual, but the meridional gradient (and therefore probably the MTF) will be higher than usual.  As stated in the text: "For instance, if the large-scale flow is less efficient than usual at advecting heat poleward (perhaps due to the orientation or intensity of the main current), mesoscale variability may take up more of this flux instead." (line numbers 492-494)

Equation (13): I think this should be sqrt(2EKE), since EKE=1/2(u^2+v^2), but for the question of correlation it doesn't matter

Indeed, we are only aiming to show proportionality rather than equivalence in equation (13), and as noted the factor of 2 or sqrt(2) will not affect the correlation or fraction of variance explained. It is now stated explicitly in the text that equation (13) is only a statement of proportionality. (line number 517)

Line 340: Mixing length and background gradient vary in space and time. A shift of the GS axis in the North Atlantic for example completely changes the locations of the major temperature gradients.

The reviewer is correct of course, and the variation in mixing lengths and gradients is a likely cause of the poor correlation of EKE and MTF in energetic western boundary current regions. We have attempted to reduce the impact of the exact location of the jet axis on the EKE-MTF correlation, by using zonal smoothers on EKE and MTF.  This is explained more clearly in the revision: "In this analysis, the zonal smoothing filter used on MTF in previous figures is applied to both EKE and MTF in order to reduce the impact of shifts in the location of currents and focus on the regional relationship between EKE and MTF." (line numbers 519-521)

However, mixing length and background gradients are still more likely to vary in energetic regions such as the Gulf Stream, and the text emphasizes this possibility in the discussion: "Moreover, strong velocity jets tend to suppress diffusivity across fronts by reducing the mixing

length (Ferrari and Nikurashin, 2010); hence variations of the mixing length and background gradients unrelated to EKE interfere with the relationship between EKE and cross-frontal fluxes." (line numbers 535-537)

Line 357: local covariance between v' and T' is the definition of eddy heat flux

The distinction between *covariance* and *correlation* is important. Covariance includes the magnitude of the v' and T' variations, while correlation focuses more on the "efficiency" of these variations in producing a net flux. This has been clarified in the text: "Ultimately the MTF depends not only on the level of mesoscale velocity and temperature variation, but on the local correlation between $v_M$ and $T_M$ (Delman and Lee, 2020), which in turn is a measure of the efficiency of mesoscale motions in producing a net directional flux. The factors affecting this $v_M$-$T_M$ correlation locally need to be investigated more in future work." (line numbers 537-540)

**References:**

Delman, A., & Lee, T. (2020). A new method to assess mesoscale contributions to meridional heat transport in the North Atlantic Ocean. *Ocean Science*, *16*(4), 979–995. https://doi.org/10.5194/os-16-979-2020

Sun, B., Liu, C., & Wang, F. (2019). Global meridional eddy heat transport inferred from Argo and altimetry observations. *Scientific Reports*, *9*(1), 1–10. https://doi.org/10.1038/s41598-018-38069-2

Wekerle, C., Wang, Q., Danilov, S., Schourup-Kristensen, V., von Appen, W. J., & Jung, T. (2017). Atlantic Water in the Nordic Seas: Locally eddy-permitting ocean simulation in a global setup. *Journal of Geophysical Research: Oceans*, *122*(2), 914–940. https://doi.org/10.1002/2016JC012121

Zhao, J., Bower, A., Yang, J., & Lin, X. (2018). Meridional heat transport variability induced by mesoscale processes in the subpolar North Atlantic. *Nature Communications*, *9*(1), 1–9. https://doi.org/10.1038/s41467-018-03134-x

---

## Referee Report (RR1)

The authors did a great job streamlining the paper and addressing concerns raised by the reviewers. I find it much easier to follow now and I think the paper can be published as it is, if the following very minor concern about the mixing length equations is addressed:

- Equation (12) $MTF \approx -\kappa \frac{\partial T}{\partial y}$

And remove the "which is generally negative or downgradient" (line 347). Diffusivity is per definition positive, a negative value of $\kappa$ would indicate something going from a diffuse state to a concentrated state, which physically makes no sense. The negative sign "belongs" to the temperature gradient, indicating that the flux is downgradient.

- Equation (13) $\kappa \propto \sqrt{v'^2} \, L_{mix} \propto \sqrt{EKE} \, L_{mix}$

Since $\sqrt{v'^2} \neq \sqrt{EKE}$, but they are proportional with a factor of $\sqrt{2}$.

One major concern raised by reviewer#1 and me was about the underestimation of EKE in the model. I think the Appendix B addresses this problem adequately regarding the analysis of this paper and shows that the results are not overly sensitive to the low EKE. However, for the future it might be reasonable to analyze why the POP2 model (contrary to other ocean models with comparable horizontal resolution) underestimates EKE so substantially and to what extend this influences results from analyzing output of this specific ocean model.

---

## Author Response (AR2)

**Response to referee #2, 2nd round:**

Overall the paper has much improved. I think it is fine to publish this content.

Thanks very much to the reviewer for their re-review and comments. Our responses and corresponding revisions to the text are given below in red text. Line numbers refer to the "tracked changes" version of the manuscript, and differ from the version without tracked changes.

Some comments:

L20 – "Along frontal gradients" is jargon that I would not use to open a paper with. Maybe rewrite this sentence and be clear what this actually is. Specially the combination along and frontal.

The first sentence of the introduction has been rewritten to use more general wording:

"In regions of the ocean where waters of different temperatures and densities converge, instabilities form that transport heat across latitude lines." (lines 20-21)

L198 – You mean the surface middle of the ocean, not the "interior" or Abyss, right? Please clarify.

Yes, this is correct. The sentence has been rewritten to be more specific:

"We note that the significant low EKE bias in the more quiescent middle regions of the ocean (Figure 1) implies that MTF may be underestimated in these regions, especially at low latitudes." (lines 201-202)

L255 – Maybe tone down this statement. Mesoscale is locally important perhaps, but it seems overrated here.

We see no reason to tone down the statement given the results that are shown in Figure 7. The mesoscale contribution to meridional HT is of comparable magnitude to the overturning and large-scale contributions near 40° latitude in both hemispheres. The statement has been slightly revised to specify the latitude range that is being discussed:

"This suggests that the mesoscale plays an important part in conveying poleward meridional HT from the subtropical to subpolar gyres in the Northern Hemisphere near 40°N, and across the equatorward edges of the Southern Ocean near 40°S." (lines 257-259)

L345 The diffusivity is positive! It can operate on downgradient fluxes, hence the minus sign.

Since it is more commonly used, we will adopt the convention where the diffusivity is positive in most cases (i.e., when there is a downgradient flux). Equations (12)-(13) and the text have been changed accordingly.

L345 – This is very rough, as you are ignoring the east-west velocity. How does this affect the conclusions in this paragraph? These are important in many regions.

There is an assumption implicit in equation (13) that the standard deviations of the zonal and meridional velocity are comparable; if so then $\sqrt{v'^2} \approx \sqrt{\frac{1}{2}(u'^2 + v'^2)} = \sqrt{EKE}$. This assumption has been stated explicitly in the revised text:

"(The approximation in eq. 13 assumes that $u'^2 \approx v'^2$, which is valid where dynamics are generally isotropic, but not where flows are strongly asymmetric and nearly aligned with the zonal or meridional axis.)" (lines 351-352)

L365 – If you want to address the effect of eddies this way, perhaps it is better to use the maps of diffusivities that include suppression effects? Examples are Klocker and Abernathey 2014 or Groeskamp et al (2020). Here MLT is used, together with suppression and this could be a better proxy.

The reviewer makes a good point, that maps of the suppression factor might also provide insight as to where suppression effects would interfere with the EKE-MTF relationship. A mention of this has been added to the text:

"Moreover, strong velocity jets tend to suppress diffusivity across fronts by reducing the mixing length (Ferrari and Nikurashin, 2010); the suppression effect of the flow is pronounced in many of the same energetic regions where EKE is a poor proxy for MTF (Klocker and Abernathey, 2014; Groeskamp et al., 2020)." (lines 369-372)

L389 – "The effects of these recirculations are not included in a temporally-defined eddy flux, but these recirculations still need to be parameterized as they will not be accurately simulated in a coarse-resolution" I'm not sure if you can make this statement with such certainty based on what we have seen. Maybe you have presented the arguments, if so, I suggest to provide them here with exact references to figures and section in this paper, and preferably other literature. I'm sure this is not the first time this is mentioned. This, because it is a big statement to make. If correct, potentially very important. Which would be a nice outcome of the paper.

This is in fact a major conclusion of the paper, and we think that the evidence from Figures 5 and 6 supports the conclusion that it is essential to simulate mesoscale structures to accurately

represent basin-scale (and global) meridional heat transports. We have stated this more specifically in the conclusion section of the paper.

"Moreover, stationary mesoscale recirculations associated with western boundary currents contribute substantially to mesoscale meridional HT (e.g., the Kuroshio and Agulhas as shown in Figure 5). The effects of stationary recirculations are not included in a temporally-defined eddy flux, as these recirculations are part of the time-mean circulation. However, a coarse-resolution model that does not accurately simulate temperature and velocity gradients at western boundaries (Figure 6) may neglect contributions to the meridional HT that are >0.1 PW, even if the model accurately represents the large-scale flow and mass transport. Therefore, the effects of these stationary recirculations still need to be parameterized just as the effects of transient eddies are parameterized." (lines 393-400)

**Response to referee #3, 2$^{nd}$ round:**

Thanks very much to the reviewer for their re-review and comments. Our responses and corresponding revisions to the text are given below in red text. Line numbers refer to the "tracked changes" version of the manuscript, and differ from the version without tracked changes.

The authors did a great job streamlining the paper and addressing concerns raised by the reviewers. I find
it much easier to follow now and I think the paper can be published as it is, if the following very minor
concern about the mixing length equations is addressed:
- Equation (12) $MTF \approx - \kappa(\partial T/\partial y)$

And remove the "which is generally negative or downgradient" (line 347). Diffusivity is per definition positive, a negative value of $\kappa$ would indicate something going from a diffuse state to a
concentrated state, which physically makes no sense. The negative sign "belongs" to the temperature
gradient, indicating that the flux is downgradient.

Equations (12)-(13) have been changed so that the diffusivity is positive when there is a downgradient flux.

- Equation (13) $\kappa \propto \sqrt{v'2} \ Lmix \propto \sqrt{EKE} \ Lmix$
Since $\sqrt{v'2} \neq \sqrt{EKE}$, but they are proportional with a factor of $\sqrt{2}$.

The factor of $\sqrt{2}$ is actually not necessary; if the assumption is made that the variances of $u'^2$ and $v'^2$ are comparable (as they would be in generally isotropic turbulence), then $\sqrt{v'^2} \approx \sqrt{\frac{1}{2}(u'^2 + v'^2)} = \sqrt{EKE}$. This assumption has been explicitly stated in the revised text.

"(The approximation in eq. 13 assumes that $u'^2 \approx v'^2$, which is valid where dynamics are generally isotropic, but not where flows are strongly asymmetric and nearly aligned with the zonal or meridional axis.)" (lines 351-352)

One major concern raised by reviewer#1 and me was about the underestimation of EKE in the model. I think the Appendix B addresses this problem adequately regarding the analysis of this paper and shows that the results are not overly sensitive to the low EKE. However, for the future it might be reasonable to analyze why the POP2 model (contrary to other ocean models with comparable horizontal resolution) underestimates EKE so substantially and to what extend this influences results from analyzing output of this specific ocean model.

We agree that this is an important question, though beyond the scope of our paper, and there is no clear answer known to us. Other high-resolution "eddy-permitting" models also have a low EKE bias, though the bias in this POP2 simulation is particularly pronounced. The bias is fairly pervasive in the middle of ocean basins, but generally limited to areas where mesoscale energy is low or moderate (quiescent regions). Some of the bias can be attributed to the misplacement of currents (e.g., the Azores Current, the absence of the Northwest Corner in the North Atlantic Current, the Azores Current). Another possible reason is that grid-scale (biharmonic) viscosity and diffusivity values are set for optimum behavior in energetic current regions, but that these values are not ideal for more quiescent regions and tend to suppress instability development too much. As stated in Section 2.1:

"One possible explanation is that the biharmonic viscosity $\nu_0 = -2.7 \times 10^{10}$ m$^4$ s$^{-1}$ and diffusivity $\kappa_0 = -3 \times 10^9$ m$^4$ s$^{-1}$ values used in this model simulation are well tuned for energetic regions such as western boundary currents, but may suppress too much mesoscale activity in the less energetic ocean interiors. However, the low bias in time-mean EKE is not unique to this eddy-permitting model (e.g., Volkov et al., 2008, 2010; Tréguier et al., 2012)." (lines 95-99)